# DisDiff: Unsupervised Disentanglement of Diffusion Probabilistic Models

**Tao Yang**[1][*], **Yuwang Wang**[2][†], **Yan Lu**[3] , **Nanning Zheng**[1][†]
yt14212@stu.xjtu.edu.cn,
[1]National Key Laboratory of Human-Machine Hybrid Augmented Intelligence,
National Engineering Research Center for Visual Information and Applications,
and Institute of Artificial Intelligence and Robotics,
Xi'an Jiaotong University,
[2]Tsinghua University, Shanghai AI Laboratory,
[3]Microsoft Research Asia
https://github.com/thomasmry/DisDiff

## Abstract

Targeting to understand the underlying explainable factors behind observations and modeling the conditional generation process on these factors, we connect disentangled representation learning to Diffusion Probabilistic Models (DPMs) to take advantage of the remarkable modeling ability of DPMs. We propose a new task, disentanglement of (DPMs): given a pre-trained DPM, without any annotations of the factors, the task is to automatically discover the inherent factors behind the observations and disentangle the gradient fields of DPM into sub-gradient fields, each conditioned on the representation of each discovered factor. With disentangled DPMs, those inherent factors can be automatically discovered, explicitly represented, and clearly injected into the diffusion process via the sub-gradient fields. To tackle this task, we devise an unsupervised approach named DisDiff, achieving disentangled representation learning in the framework of DPMs. Extensive experiments on synthetic and real-world datasets demonstrate the effectiveness of DisDiff.

## 1 Introduction

As one of the most successful generative models, diffusion probabilistic models (DPMs) achieve remarkable performance in image synthesis. They use a series of probabilistic distributions to corrupt images in the forward process and train a sequence of probabilistic models converging to image distribution to reverse the forward process. Despite the remarkable success of DPM in tasks such as image generation [41], text-to-images [37], and image editing [32], little attention has been paid to representation learning [49] based on DPM. Diff-AE [33] and PDAE [49] are recently proposed methods for representation learning by reconstructing the images in the DPM framework. However, the learned latent representation can only be interpreted relying on an extra pre-trained with predefined semantics linear classifier. In this paper, we will refer DPMs exclusively to the Denoising Diffusion Probabilistic Models (DDPMs) [19].

On the other hand, disentangled representation learning [17] aims to learn the representation of the underlying explainable factors behind the observed data and is thought to be one of the possible ways for AI to understand the world fundamentally. Different factors correspond to different kinds of image variations, respectively and independently. Most of the methods learn the disentangled representation

---

[*]Work done during internships at Microsoft Research Asia.
[†]Corresponding authors: wang-yuwang@mail.tsinghua.edu.cn; nnzheng@mail.xjtu.edu.cn.

37th Conference on Neural Information Processing Systems (NeurIPS 2023).

based on generative models, such as VAE [16, 5, 23] and GAN [27]. The VAE-based methods have an inherent trade-off between the disentangling ability and generating quality [16, 5, 23]. The GAN-based methods suffer from the problem of reconstruction due to the difficulty of gan-inversion [44].

In this paper, we connect DPM to disentangled representation learning and propose a new task: the disentanglement of DPM. Given a pre-trained DPM model, the goal of disentanglement of DPM is to learn disentangled representations for the underlying factors in an unsupervised manner, and learn the corresponding disentangled conditional sub-gradient fields, with each conditioned on the representation of each discovered factor, as shown in Figure 1.

The benefits of the disentanglement of DPM are two-fold: $(i)$ It enables totally unsupervised controlling of images by automatically discovering the inherent semantic factors behind the image data. These factors help to extend the DPM conditions information from human-defined ones such as annotations [49]/image-text pairs [21], or supervised pre-trained models [22] such as CLIP [34]. One can also flexibly sample partial conditions on the part of the information introduced by the superposition of the sub-gradient field, which is novel in existing DPM works. $(ii)$ DPM has remarkable performance on image generation quality and is naturally friendly for the inverse problem, e.g., the inversion of DDIM [39], PDAE. Compared to VAE (trade-off between the disentangling ability and generating quality) or GAN (problem of gan-inversion), DPM is a better framework for disentangled representation learning. Besides, as Locatello et al. [30] points out, other inductive biases should be proposed except for total correlation. DPM makes adopting constraints from all different timesteps possible as a new type of inductive bias. Further, as Srivastava et al. [42] points out, and the data information includes: factorized and non-factorized. DPM has the ability to sample non-factorized (non-conditioned) information [18], which is naturally fitting for disentanglement.

To address the task of disentangling the DPM, we further propose an unsupervised solution for the disentanglement of a pre-trained DPM named DisDiff. DisDiff adopts an encoder to learn the disentangled presentation for each factor and a decoder to learn the corresponding disentangled conditional sub-gradient fields. We further propose a novel Disentangling Loss to make the encoded representation satisfy the disentanglement requirement and reconstruct the input image.

Our main contributions can be summarized as follows:

- We present a new task: disentanglement of DPM, disentangling a DPM into several disentangled sub-gradient fields, which can improve the interpretability of DPM.

- We build an unsupervised framework for the disentanglement of DPM, DisDiff, which learns a disentangled representation and a disentangled gradient field for each factor.

- We propose enforcing constraints on representation through the diffusion model's classifier guidance and score-based conditioning trick.

## 2 Related Works

**Diffusion Probabilistic Models** DPMs have achieved comparable or superior image generation quality [38, 40, 19, 41, 20] than GAN [13]. Diffusion-based image editing has drawn much attention, and there are mainly two categories of works. Firstly, image-guided works edit an image by mixing the latent variables of DPM and the input image [7, 31, 32]. However, using images to specify the attributes for editing may cause ambiguity, as pointed out by Kwon et al. [25]. Secondly, the classifier-guided works [9, 1, 28] edit images by utilizing the gradient of an extra classifier. These methods require calculating the gradient, which is costly. Meanwhile, these methods require annotations or models pre-trained with labeled data. In this paper, we propose DisDiff to edit the image in an unsupervised way. On the other hand, little attention has been paid to representation learning in the literature on the diffusion model. Two related works are Diff-ae [33] and PDAE [49]. Diff-ae [33] proposes a diffusion-based auto-encoder for image reconstruction. PDAE [49] uses a pre-trained DPM to build an auto-encoder for image reconstruction. However, the latent representation learned by these two works does not respond explicitly to the underlying factors of the dataset.

**Disentangled Representation Learning** Bengio et al. [2] introduced disentangled representation learning. The target of disentangled representation learning is to discover the underlying explanatory factors of the observed data. The disentangled representation is defined as each dimension of the representation corresponding to an independent factor. Based on such a definition, some VAE-based

works achieve disentanglement [5, 23, 16, 3] only by the constraints on probabilistic distributions of representations. Locatello et al. [30] point out the identifiable problem by proving that only these constraints are insufficient for disentanglement and that extra inductive bias is required. For example, Yang et al. [47] propose to use symmetry properties modeled by group theory as inductive bias. Most of the methods of disentanglement are based on VAE. Some works based on GAN, including leveraging a pre-trained generative model [36]. Our DisDiff introduces the constraint of all time steps during the diffusion process as a new type of inductive bias. Furthermore, DPM can sample non-factorized (non-conditioned) information [18], naturally fitting for disentanglement. In this way, we achieve disentanglement for DPMs.

## 3 Background

### 3.1 Diffusion Probabilistic Models (DPM)

We take DDPM [19] as an example. DDPM adopts a sequence of fixed variance distributions $q(x_t|x_{t-1})$ as the forward process to collapse the image distribution $p(x_0)$ to $\mathcal{N}(0, I)$. These distributions are

$$q(x_t|x_{t-1}) = \mathcal{N}(x_t; \sqrt{1-\beta_t}x_{t-1}, \beta_t I). \tag{1}$$

We can then sample $x_t$ using the following formula $x_t \sim \mathcal{N}(x_t; \sqrt{\bar{\alpha}_t}x_0, (1-\bar{\alpha}_t))$, where $\alpha_t = 1-\beta_t$ and $\bar{\alpha}_t = \Pi_{t=1}^t \alpha_t$, i.e., $x_t = \sqrt{\bar{\alpha}_t}x_0 + \sqrt{1-\bar{\alpha}_t}\epsilon$. The reverse process is fitted by using a sequence of Gaussian distributions parameterized by $\theta$:

$$p_\theta(x_{t-1}|x_t) = \mathcal{N}(x_t; \mu_\theta(x_t, t), \sigma_t I), \tag{2}$$

where $\mu_\theta(x_t, t)$ is parameterized by an Unet $\epsilon_\theta(x_t, t)$, it is trained by minimizing the variational upper bound of negative log-likelihood through:

$$\mathcal{L}_\theta = \mathop{\mathbb{E}}_{x_0, t, \epsilon} \|\epsilon - \epsilon_\theta(x_t, t)\|. \tag{3}$$

### 3.2 Representation learning from DPMs

The classifier-guided method [9] uses the gradient of a pre-trained classifier, $\nabla_{x_t} \log p(y|x_t)$, to impose a condition on a pre-trained DPM and obtain a new conditional DPM: $\mathcal{N}(x_t; \mu_\theta(x_t, t) + \sigma_t \nabla_{x_t} \log p(y|x_t), \sigma_t)$. Based on the classifier-guided sampling method, PDAE [49] proposes an approach for pre-trained DPM by incorporating an auto-encoder. Specifically, given a pre-trained DPM, PDAE introduces an encoder $E_\phi$ to derive the representation by equation $z = E_\phi(x_0)$. They use a decoder estimator $G_\psi(x_t, z, t)$ to simulate the gradient field $\nabla_{x_t} \log p(z|x_t)$ for reconstructing the input image.

By this means, they create a new conditional DPM by assembling the unconditional DPM as the decoder estimator. Similar to regular DPM $\mathcal{N}(x_t; \mu_\theta(x_t, t) + \sigma_t G_\psi(x_t, z, t), \sigma_t)$, we can use the following objective to train encoder $E_\phi$ and the decoder estimator $G_\psi$:

$$\mathcal{L}_\psi = \mathop{\mathbb{E}}_{x_0, t, \epsilon} \|\epsilon - \epsilon_\theta(x_t, t) + \frac{\sqrt{\alpha_t}\sqrt{1-\bar{\alpha}_t}}{\beta_t}\sigma_t G_\psi(x_t, z, t)\|. \tag{4}$$

## 4 Method

In this section, we initially present the formulation of the proposed task, Disentangling DPMs, in Section 4.1. Subsequently, an overview of DisDiff is provided in Section 4.2. We then elaborate on the detailed implementation of the proposed Disentangling Loss in Section 4.3, followed by a discussion on balancing it with the reconstruction loss in Section 4.4.

### 4.1 Disentanglement of DPM

We assume that dataset $\mathcal{D} = \{x_0|x_0 \sim p(x_0)\}$ is generated by $N$ underlying ground truth factors $f^c$, where each $c \in \mathcal{C} = \{1, \ldots, N\}$, with the data distribution $p(x_0)$. This implies that the underlying factors condition each sample. Moreover, each factor $f^c$ follows the distribution $p(f^c)$, where $p(f^c)$ denotes the distribution of factor $c$. Using the Shapes3D dataset as an example, the underlying

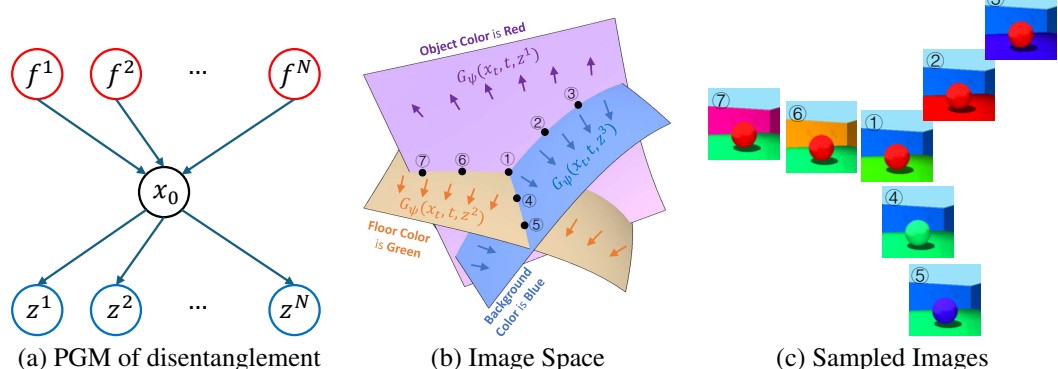

| (a) PGM of disentanglement | (b) Image Space | (c) Sampled Images |

Figure 1: Illustration of disentanglement of DPMs. (a) is the diagram of Probabilistic Graphical Models (PGM). (b) is the diagram of image space. (c) is the demonstration of sampled images in (b). Surface indicates the conditional data distribution of a single factor $p(x|f^c)$. Different colors correspond to various factors. Here, we show three factors: object color, background color, and floor color. Arrows are gradient fields $\nabla_{x_t} \log p(f^c|x_t)$ modeled by using a decoder $G_\phi^c(x_t, t, z^c)$. The black points are the sampled images shown in (b).

concept factors include background color, floor color, object color, object shape, object scale, and pose. We illustrate three factors (background color, floor color, and object color) in Figure 1 (c). The conditional distributions of the factors are $\{p(x_0|f^c)|c \in \mathcal{C}\}$, each of which can be shown as a curved surface in Figure 1 (b). The relation between the conditional distribution and data distribution can be formulated as: The conditional distributions of the factors, denoted as $\{p(x_0|f^c)|c \in \mathcal{C}\}$, can each be represented as a curved surface in Figure 1 (b). The relationship between the conditional distribution and the data distribution can be expressed as follows:

$$p(x_0) = \int p(x_0|f^c)p(f^c)df^c = \int \cdots \int p(x_0|f^1, \ldots, f^N)p(f^1) \ldots p(f^N)df^1 \ldots df^N, \quad (5)$$

this relationship holds true because $f^1, \ldots, f^N$ and $x_0$ form a v-structure in Probabilistic Graphical Models (PGM), as Figure 1 (a) depicts. A DPM learns a model $\epsilon_\theta(x_t, t)$, parameterized by $\theta$, to predict the noise added to a sample $x_t$. This can then be utilized to derive a score function: $\nabla_{x_t} \log p(x_t) = 1/\sqrt{\alpha_t - 1} \cdot \epsilon_\theta(x_t, t)$. Following [9] and using Bayes' Rule, we can express the score function of the conditional distribution as:

$$\nabla_{x_t} \log p(x_t|f^c) = \nabla_{x_t} \log p(f^c|x_t) + \nabla_{x_t} \log p(x_t), \quad (6)$$

By employing the score function of $p(x_t|f^c)$, we can sample data conditioned on the factor $f^c$, as demonstrated in Figure 1 (b). Furthermore, the equation above can be extended to one conditioned on a subset $\mathcal{S} \subseteq \mathcal{C}$ by substituting $f^c$ with $\mathcal{S}$. The goal of disentangling a DPM is to model $\nabla_{x_t} \log p(x_t|f^c)$ for each factor $c$. Based on the equation above, we can learn $\nabla_{x_t} \log p(f^c|x_t)$ for a pre-trained DPM instead, corresponding to the arrows pointing towards the curve surface in Figure 1 (b). However, in an unsupervised setting, $f^c$ is unknown. Fortunately, we can employ disentangled representation learning to obtain a set of representations $\{z^c|c = 1, \ldots, N\}$. There are two requirements for these representations: $(i)$ they must encompass all information of $x_0$, which we refer to as the *completeness* requirement, and $(ii)$ there is a bijection, $z^c \mapsto f^c$, must exist for each factor $c$, which we designate as the *disentanglement* requirement. Utilizing these representations, the gradient $\nabla_{x_t} \log p(z^c|x_t)$ can serve as an approximation for $\nabla_{x_t} \log p(f^c|x_t)$. In this paper, we propose a method named DisDiff as a solution for the disentanglement of a DPM.

## 4.2 Overview of DisDiff

The overview framework of DisDiff is illustrated in Figure 2. Given a pre-trained unconditional DPM on dataset $\mathcal{D}$ with factors $\mathcal{C}$, e.g., a DDPM model with parameters $\theta$, $p_\theta(x_{t-1}|x_t) = \mathcal{N}(x_{t-1}; \mu_\theta(x_t, t), \sigma_t)$, our target is to disentangle the DPM in an unsupervised manner. Given an input $x_0 \in \mathcal{D}$, for each factor $c \in \mathcal{C}$, our goal is to learn the disentangled representation $z^c$ simultaneously and its corresponding disentangled gradient field $\nabla_{x_t} \log p(z^c|x_t)$. Specifically, for each representation $z^c$, we employ an encoder $E_\phi$ with learnable parameters $\phi$ to obtain the

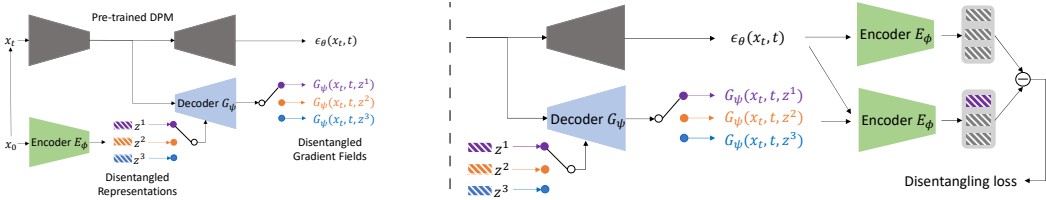

(a) The networks architecture of DisDiff    (b) The demonstration of disentangling loss

Figure 2: Illustration of DisDiff. (a) Grey networks indicate the pre-trained Unet of DPM $\epsilon_\theta(x_t, t)$. Image $x_0$ is first encoded to representations $\{z^1, z^2, \ldots z^N\}$ of different factors by encoder $E_\phi$ ($N = 3$ in the figure), which are then decoded by decoder $G_\psi$ to obtain the gradient field of the corresponding factor. We can sample the image under the corresponding condition with the obtained gradient field. (b) We first sample a factor $c$ and decode the representation $z^c$ to obtain the gradient field of the corresponding factor, which allows us to obtain the predicted $x_0$ of that factor. At the same time, we can obtain the predicted $\hat{x}_0$ of the original pre-trained DPM. We then encode the images into two different representations and calculate the disentangling loss based on them.

representations as follows: $E_\phi(x_0) = \{E_\phi^1(x_0), E_\phi^2(x_0), \ldots, E_\phi^N(x_0)\} = \{z^1, z^2, \ldots, z^N\}$. As the random variables $z^1, \ldots, z^N$ and $x_0$ form a common cause structure, it is evident that $z^1, \ldots, z^N$ are independent when conditioned on $x_0$. Additionally, this property also holds when conditioning on $\epsilon$. We subsequently formulate the gradient field, denoted as $\nabla_{x_t} \log p(z^{\mathcal{S}}|x_t)$, with $z^{\mathcal{S}} = \{z^c | c \in \mathcal{S}\}$, conditioned on a factor subset $\mathcal{S} \subseteq \mathcal{C}$, as follows:

$$\nabla_{x_t} \log p(z^S|x_t) = \sum_{c \in S} \nabla_{x_t} \log p(z^c|x_t). \tag{7}$$

Adopting the approach from [9, 49], we formulate the conditional reverse process as a Gaussian distribution, represented as $p_\theta(x_{t-1}|x_t, z^{\mathcal{S}})$. In conjunction with Eq. 7, this distribution exhibits a shifted mean, which can be described as follows:

$$\mathcal{N}(x_{t-1}; \mu_\theta(x_t, t) + \sigma_t \sum_{c \in \mathcal{S}} \nabla_{x_t} \log p(z^c|x_t), \sigma_t). \tag{8}$$

Directly employing $\nabla_{x_t} \log p(z^c|x_t)$ introduces computational complexity and complicates the diffusion model training pipeline, as discussed in [18]. Therefore, we utilize a network $G_\psi(x_t, z^c, t)$, with $c \in \mathcal{S}$ and parameterized by $\psi$, to estimate the gradient fields $\nabla_{x_t} \log p(z^{\mathcal{S}}|x_t)$. To achieve this goal, we first use $\sum_{c \in \mathcal{C}} G_\psi(x_t, z^c, t)$ to estimate $\nabla_{x_t} \log p(z^{\mathcal{C}}|x_t)$ based on Eq. 7. Therefore, we adopt the same loss of PDAE [49] but replace the gradient field with the summation of $G_\psi(x_t, z^c, t)$ as

$$\mathcal{L}_r = \mathbb{E}_{x_0, t, \epsilon} \left\| \epsilon - \epsilon_\theta(x_t, t) + \frac{\sqrt{\alpha_t}\sqrt{1 - \bar{\alpha}_t}}{\beta_t} \sigma_t \sum_{c \in \mathcal{C}} G_\psi(x_t, z^c, t) \right\|. \tag{9}$$

The aforementioned equation implies that $x_0$ can be reconstructed utilizing all disentangled representations when $\mathcal{S} = \mathcal{C}$, thereby satisfying the *completeness* requirement of disentanglement. Secondly, in the following, we fulfill the *disentanglement* requirement and the approximation of gradient field $\nabla_{x_t} \log p(z^c|x_t)$ with decoder $G_\psi(x_t, z^c, t)$. In the next section, we introduce a disentangling loss to address the remaining two conditions.

### 4.3 Disentangling Loss

In this section, we present the Disentangling Loss to address the *disentanglement* requirement and ensure that $G_\psi(x_t, z^c, t)$ serves as an approximation of $\nabla_{x_t} \log p(z^c|x_t)$. Given the difficulty in identifying sufficient conditions for disentanglement in an unsupervised setting, we follow the previous works in disentangled representation literature, which propose necessary conditions that are effective in practice for achieving disentanglement. For instance, the group constraint in [47] and maximizing mutual information in [27, 45] serve as examples. We propose minimizing the mutual information between $z^c$ and $z^k$, where $k \neq c$, as a necessary condition for disentangling a DPM.

Subsequently, we convert the objective of minimizing mutual information into a set of constraints applied to the representation space. We denote $\hat{x}_0$ is sampled from the pre-trained unconditioned

DPM using $\epsilon_\theta(x_t, t)$, $\hat{x}_0^c$ is conditioned on $z^c$ and sampled using $p_\theta(x_{t-1}|x_t, z^c)$. Then we can extract the representations from the samples $\hat{x}_0$ and $\hat{x}_0^c$ with $E_\phi$ as $\hat{z}^k = E_\phi^k(\hat{x}_0)$ and $\hat{z}^{k|c} = E_\phi^k(\hat{x}_0^c)$, respectively. According to Proposition 1, we can minimize the proposed upper bound (estimator) to minimize the mutual information.

**Proposition 1.** *For a PGM in Figure 1 (a), we can minimize the upper bound (estimator) for the mutual information between $z^c$ and $z^k$ (where $k \neq c$) by the following objective:*

$$\min_{k,c,x_0,\hat{x}_0^c} \mathbb{E} \, \|\hat{z}^{k|c} - \hat{z}^k\| - \|\hat{z}^{k|c} - z^k\| + \|\hat{z}^{c|c} - z^c\| \tag{10}$$

The proof is presented in Appendix K. We have primarily drawn upon the proofs in CLUB [6] and VAE [24] as key literature sources to prove our claims. As outlined below, we can partition the previously mentioned objectives into two distinct loss function components.

**Invariant Loss**. In order to minimize the first part, $\|\hat{z}^{k|c} - \hat{z}^k\|$, where $k \neq c$, we need to minimize $\|\hat{z}^{k|c} - \hat{z}^k\|$, requiring that the $k$-th ($k \neq c, k \in \mathcal{C}$) representation remains unchanged. We calculate the distance scalar between their $k$-th representation as follows:

$$d_k = \|\hat{z}^{k|c} - \hat{z}^k\|. \tag{11}$$

We can represent the distance between each representation using a distance vector $d = [d_1, d_2, \ldots, d_C]$. As the distances are unbounded and their direct optimization is unstable, we employ the CrossEntropy loss[3] to identify the index $c$, which minimizes the distances at other indexes. The invariant loss $\mathcal{L}_{in}$ is formulated as follows:

$$\mathcal{L}_{in} = \mathbb{E}_{c,\hat{x}_0,\hat{x}_0^c} [CrossEntropy(d, c)]. \tag{12}$$

**Variant Loss**. In order to minimize the second part, $\|\hat{z}^{c|c} - z^c\| - \|\hat{z}^{k|c} - z^k\|$, in Eq. 10. Similarly, we implement the second part in CrossEntropy loss to maximize the distances at indexes $k$ but minimize the distance at index $c$, similar to Eq. 12. Specifically, we calculate the distance scalar of the representations as shown below:

$$\begin{aligned} d_k^n &= \|\hat{z}^k - z^k\|, \\ d_k^p &= \|\hat{z}^{k|c} - z^k\|, \end{aligned} \tag{13}$$

We adopt a cross-entropy loss to achieve the subjective by minimizing the variant loss $\mathcal{L}_{va}$, we denote $[d_1^n, d_2^n, \ldots, d_N^n]$ as $d^n$ and $[d_1^p, d_2^n, \ldots, d_N^p]$ as $d^p$.

$$\mathcal{L}_{va} = \mathbb{E}_{c,x_0,\hat{x}_0,\hat{x}_0^c} [CrossEntropy(d^n - d^p, c)], \tag{14}$$

However, sampling $\hat{x}_0^c$ from $p_\theta(x_{t-1}|x_t, z^c)$ is not efficient in every training step. Therefore, we follow [9, 49] to use a score-based conditioning trick [41, 40] for fast and approximated sampling. We obtain an approximator $\epsilon_\psi$ of conditional sampling for different options of $\mathcal{S}$ by substituting the gradient field with decoder $G_\psi$:

$$\epsilon_\psi(x_t, z^{\mathcal{S}}, t) = \epsilon_\theta(x_t, t) - \sum_{c \in \mathcal{S}} \sqrt{1 - \bar{\alpha}_t} G_\psi(x_t, z^c, t). \tag{15}$$

To improve the training speed, one can achieve the denoised result by computing the posterior expectation following [8] using Tweedie's Formula:

$$\hat{x}_0^{\mathcal{S}} = \mathbb{E}[x_0|x_t, S] = \frac{x_t - \sqrt{1 - \bar{\alpha}_t}\epsilon_\psi(x_t, z^{\mathcal{S}}, t)}{\sqrt{\bar{\alpha}_t}}. \tag{16}$$

### 4.4 Total Loss

The degree of condition dependence for generated data varies among different time steps in the diffusion model [46]. For different time steps, we should use different weights for Disentangling Loss. Considering that the difference between the inputs of the encoder can reflect such changes in condition. We thus propose using the MSE distance between the inputs of the encoder as the weight coefficient:

$$\gamma_d = \lambda \|\hat{x}_0 - \hat{x}_0^c\|^2, \mathcal{L}_a = \mathcal{L}_r + \gamma_d(\mathcal{L}_{in} + \mathcal{L}_{va}). \tag{17}$$

where $\lambda$ is a hyper-parameter. We stop the gradient of $\hat{x}_0$ and $\hat{x}_0^c$ for the weight coefficient $\gamma_d$.

---

[3]We denote the function `torch.nn.CrossEntropyLoss` as CrossEntropy.

Table 1: Comparisons of disentanglement on the FactorVAE score and DCI disentanglement metrics (mean $\pm$ std, higher is better). DisDiff achieves state-of-the-art performance with a large margin in almost all the cases compared to all baselines, especially on the MPI3D dataset.

| Method | Cars3D | | Shapes3D | | MPI3D | |
|---|---|---|---|---|---|---|
| | FactorVAE score | DCI | FactorVAE score | DCI | FactorVAE score | DCI |
| *VAE-based:* | | | | | | |
| FactorVAE | $0.906 \pm 0.052$ | $0.161 \pm 0.019$ | $0.840 \pm 0.066$ | $0.611 \pm 0.082$ | $0.152 \pm 0.025$ | $0.240 \pm 0.051$ |
| $\beta$-TCVAE | $0.855 \pm 0.082$ | $0.140 \pm 0.019$ | $0.873 \pm 0.074$ | $0.613 \pm 0.114$ | $0.179 \pm 0.017$ | $0.237 \pm 0.056$ |
| *GAN-based:* | | | | | | |
| InfoGAN-CR | $0.411 \pm 0.013$ | $0.020 \pm 0.011$ | $0.587 \pm 0.058$ | $0.478 \pm 0.055$ | $0.439 \pm 0.061$ | $0.241 \pm 0.075$ |
| *Pre-trained GAN-based:* | | | | | | |
| LD | $0.852 \pm 0.039$ | $0.216 \pm 0.072$ | $0.805 \pm 0.064$ | $0.380 \pm 0.062$ | $0.391 \pm 0.039$ | $0.196 \pm 0.038$ |
| GS | $0.932 \pm 0.018$ | $0.209 \pm 0.031$ | $0.788 \pm 0.091$ | $0.284 \pm 0.034$ | $0.465 \pm 0.036$ | $0.229 \pm 0.042$ |
| DisCo | $0.855 \pm 0.074$ | $\mathbf{0.271 \pm 0.037}$ | $0.877 \pm 0.031$ | $0.708 \pm 0.048$ | $0.371 \pm 0.030$ | $0.292 \pm 0.024$ |
| *Diffusion-based:* | | | | | | |
| DisDiff-VQ (Ours) | $\mathbf{0.976 \pm 0.018}$ | $0.232 \pm 0.019$ | $\mathbf{0.902 \pm 0.043}$ | $\mathbf{0.723 \pm 0.013}$ | $\mathbf{0.617 \pm 0.070}$ | $\mathbf{0.337 \pm 0.057}$ |

# 5 Experiments

## 5.1 Experimental Setup

**Implementation Details**. $x_0$ can be a sample in an image space or a latent space of images. We take pre-trained DDIM as the DPM (DisDiff-IM) for image diffusion. For latent diffusion, we can take the pre-trained KL-version latent diffusion model (LDM) or VQ-version LDM as DPM (DisDiff-KL and DisDiff-VQ). For details of network $G_\theta$, we follow Zhang et al. [49] to use the extended Group Normalization [9] by applying the scaling & shifting operation twice. The difference is that we use learn-able position embedding to indicate $c$:

$$AdaGN(h, t, z^c) = z_s^c(t_s^c GN(h) + t_b^c) + z_b^c, \tag{18}$$

where $GN$ denotes group normalization, and $[t_s^c, t_b^c], [z_s^c, z_b^c]$ are obtained from a linear projection: $z_s^c, z_b^c = linearProj(z^c)$, $t_s^c, t_b^c = linearProj([t, p^c])$. $p^c$ is the learnable positional embedding. $h$ is the feature map of Unet. For more details, please refer to Appendix A and B.

**Datasets** To evaluate disentanglement, we follow Ren et al. [36] to use popular public datasets: Shapes3D [23], a dataset of 3D shapes. MPI3D [12], a 3D dataset recorded in a controlled environment, and Cars3D [35], a dataset of CAD models generated by color renderings. All experiments are conducted on 64x64 image resolution, the same as the literature. For real-world datasets, we conduct our experiments on CelebA [29].

**Baselines & Metrics** We compare the performance with VAE-based and GAN-based baselines. Our experiment is conducted following DisCo. All of the baselines and our method use the same dimension of the representation vectors, which is 32, as referred to in DisCo. Specifically, the VAE-based models include FactorVAE [23] and $\beta$-TCVAE [5]. The GAN-based baselines include InfoGAN-CR [27], GANspace (GS) [14], LatentDiscovery (LD) [43] and DisCo [36]. Considering the

Table 2: Ablation study of DisDiff on image encoder, components, batchsize and token numbers on Shapes3D.

| Method | FactorVAE score | DCI |
|---|---|---|
| DisDiff-IM | 0.783 | 0.655 |
| DisDiff-KL | 0.837 | 0.660 |
| DisDiff-VQ | 0.902 | 0.723 |
| DisDiff-VQ wo $\mathcal{L}_{in}$ | 0.782 | 0.538 |
| DisDiff-VQ wo $\mathcal{L}_{va}$ | 0.810 | 0.620 |
| DisDiff-VQ wo $\mathcal{L}_{dis}$ | 0.653 | 0.414 |
| wo detach | 0.324 | 0.026 |
| constant weighting | 0.679 | 0.426 |
| loss weighting | 0.678 | 0.465 |
| attention condition | 0.824 | 0.591 |
| wo pos embedding | 0.854 | 0.678 |
| wo orth embedding | 0.807 | 0.610 |
| latent number $N=6$ | 0.865 | 0.654 |
| latent number $N=10$ | 0.902 | 0.723 |

influence of performance on the random seed. We have ten runs for each method. We use two representative metrics: the FactorVAE score [23] and the DCI [11]. However, since $\{z^c | c \in \mathcal{C}\}$ are vector-wise representations, we follow Du et al. [10] to perform PCA as post-processing on the representation before evaluation.

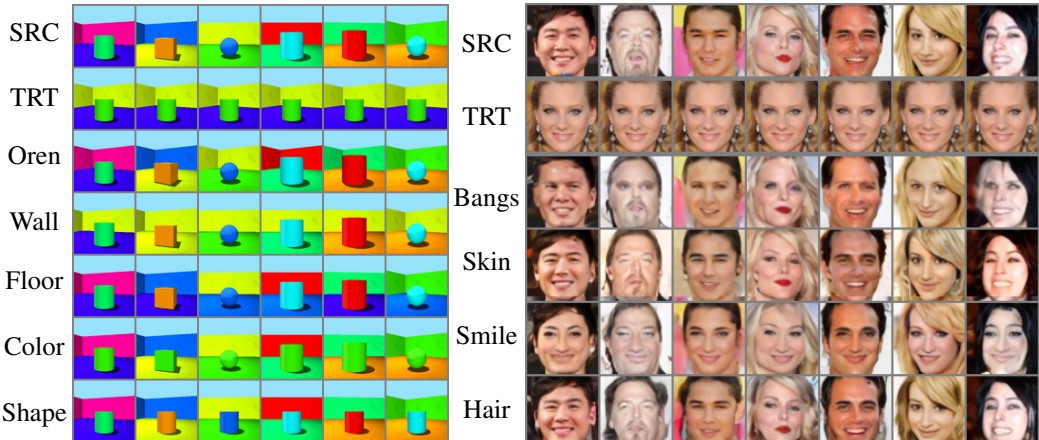

Figure 3: The qualitative results on Shapes3D and CelebA. The source (SRC) images provide the representations of the generated image. The target (TRT) image provides the representation for swapping. Other images are generated by swapping the representation of the corresponding factor. The learned factors on Shapes3D are orientation (Oren), wall color (Wall), floor color (Floor), object color (Color), and object shape (Shape). For more visualizations, please refer to Appendix C-H.

## 5.2 Main Results

We conduct the following experiments to verify the disentanglement ability of the proposed DisDiff model. We regard the learned representation $z^c$ as the disentangled one and use the popular metrics in disentangled representation literature for evaluation. The quantitative comparison results of disentanglement under different metrics are shown in Table 1. The table shows that DisDiff outperforms the baselines, demonstrating the model's superior disentanglement ability. Compared with the VAE-based methods, since these methods suffer from the trade-off between generation and disentanglement [26], DisDiff does not. As for the GAN-based methods, the disentanglement is learned by exploring the latent space of GAN. Therefore, the performance is limited by the latent space of GAN. DisDiff leverages the gradient field of data space to learn disentanglement and does not have such limitations. In addition, DisDiff resolves the disentanglement problem into 1000 sub-problems under different time steps, which reduces the difficulty.

We also incorporate the metric for real-world data into this paper. We have adopted the TAD and FID metrics from [45] to measure the disentanglement and generation capabilities of DisDiff, respectively. As shown in Table 3, DisDiff still outperforms new baselines in comparison. It is worth mentioning that InfoDiffusion [45] has made significant improvements in both disentanglement and generation capabilities on CelebA compared to previous baselines. This has resulted in DisDiff's advantage over the baselines on CelebA being less pronounced than on Shapes3D.

Table 3: Comparisons of disentanglement on Real-world dataset CelebA with TAD and FID metrics (mean ± std). DisDiff still achieves state-of-the-art performance compared to all baselines.

| Model | TAD | FID |
|---|---|---|
| $\beta$-VAE | $0.088 \pm 0.043$ | $99.8 \pm 2.4$ |
| InfoVAE | $0.000 \pm 0.000$ | $77.8 \pm 1.6$ |
| Diffae | $0.155 \pm 0.010$ | $22.7 \pm 2.1$ |
| InfoDiffusion | $0.299 \pm 0.006$ | $23.6 \pm 1.3$ |
| DisDiff (ours) | $\mathbf{0.3054 \pm 0.010}$ | $\mathbf{18.249 \pm 2.1}$ |

The results are taken from Table 2 in [45], except for the results of DisDiff. For TAD, we follow the method used in [45] to compute the metric. We also evaluate the metric for 5-folds. For FID, we follow Wang et al. [45] to compute the metric with five random sample sets of 10,000 images. Specifically, we use the official implementation of [48] to compute TAD, and the official GitHub repository of [15] to compute FID.

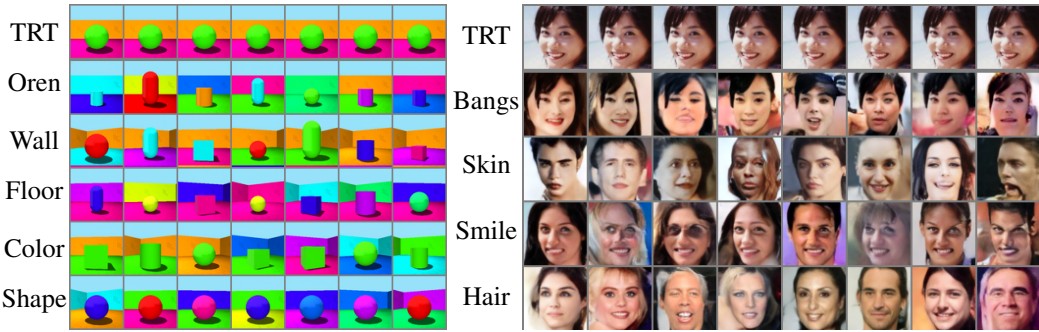

Figure 4: The partial condition sampling on Shapes3D and CelebA. The target (TRT) image provides the representation of partial sampling images. Images in each row are generated by imposing a single gradient field of the corresponding factor on the pre-trained DPM. DisDiff samples image condition on only a single factor. The sampled image has a fixed factor, e.g., the images in the third row have the same background color as the target one. The conditioned factors on Shapes3D are orientation (Oren), wall color (Wall), floor color (Floor), object color (Color), and object shape (Shape).

## 5.3 Qualitative Results

In order to analyze the disentanglement of DisDiff qualitatively, we swap the representation $z^c$ of two images one by one and sample the image conditioned on the swapped representation. We follow LDM-VQ to sample images in 200 steps. For the popular dataset of disentanglement literature, we take Shapes3D as an example. As Figure 3 shows, DisDiff successfully learned pure factors. Compared with the VAE-based methods, DisDiff has better image quality. Since there are no ground truth factors for the real-world dataset, we take CelebA as an example. As demonstrated in Figure 3, DisDiff also achieves good disentanglement on real-world datasets. Please note that, unlike DisCo [36], DisDiff can reconstruct the input image, which is unavailable for DisCo. Inspired by Kwon et al. [25], we generalize DisDiff to complex large-natural datasets FFHQ and CelebA-HQ. The experiment results are given in Appendix E-H.

## 5.4 Ablation Study

In order to analyze the effectiveness of the proposed parts of DisDiff, we design an ablation study from the following five aspects: DPM type, Disentangling Loss, loss weighting, condition type, and latent number. We take Shapes3D as the dataset to conduct these ablation studies.

**DPM type** The decomposition of the gradient field of the diffusion model derives the disentanglement of DisDiff. Therefore, the diffusion space influences the performance of DisDiff. We take Shapes3D as an example. It is hard for the model to learn shape and scale in image space, but it is much easier in the latent space of the auto-encoder. Therefore, we compare the performance of DisDiff with different diffusion types: image diffusion model, e.g., DDIM (DisDiff-IM), KL-version latent diffusion model (DisDiff-KL), and VQ-version latent diffusion model, e.g., VQ-LDM (DisDiff-VQ). As shown in Table 2, the LDM-version DisDiff outperforms the image-version DisDiff as expected.

**Disentangling Loss** Disentangling loss comprises two parts: Invariant loss $\mathcal{L}_{in}$ and Variant loss $\mathcal{L}_{va}$. To verify the effectiveness of each part, we train DisDiff-VQ without it. $\mathcal{L}_{in}$ encourages in-variance of representation not being sampled, which means that the sampled factor will not affect the representation of other factors ($z^k, k \neq c$ of generated $\hat{x}_0^c$). On the other hand, $\mathcal{L}_{va}$ encourages the representation of the sampled factor ($z^c$ of generated $\hat{x}_0^c$) to be close to the corresponding one of $x_0$. As shown in Table 2, mainly $\mathcal{L}_{in}$ promotes the disentanglement, and $\mathcal{L}_{va}$ further constrains the model and improves the performance. Note that the disentangling loss is optimized w.r.t. $G_\theta$ but not $E_\theta^c$. If the loss is optimized on both modules, as shown in Table 2, DisDiff fails to achieve disentanglement. The reason could be that the disentangling loss influenced the training of the encoder and failed to do reconstruction.

**Loss weighting** As introduced, considering that the condition varies among time steps, we adopt the difference of the encoder as the weight coefficient. In this section, we explore other options to verify its effectiveness. We offer two different weighting types: constant weighting and loss weighting. The

first type is the transitional way of weighting. The second one is to balance the Distangling Loss and diffusion loss. From Table 2, these two types of weighting hurt the performance to a different extent.

**Condition type & Latent number** DisDiff follows PDAE [49] and Dhariwal and Nichol [9] to adopt AdaGN for injecting the condition. However, there is another option in the literature: cross-attention. As shown in Table 2, cross-attention hurt the performance but not much. The reason may be that the condition is only a single token, which limits the ability of attention layers. We use learnable orthogonal positional embedding to indicate different factors. As shown in Table 2, no matter whether no positional embedding (wo pos embedding) or traditional learnable positional embedding (wo orth embedding) hurt the performance. The reason is that the orthogonal embedding is always different from each other in all training steps. The number of latent is an important hyper-parameter set in advance. As shown in Table 2, if the latent number is greater than the number of factors in the dataset, the latent number only has limited influence on the performance.

## 5.5 Partial Condition Sampling

As discussed in Section 4.2, DisDiff can partially sample conditions on the part of the factors. Specifically, we can use Eq. 15 to sample image condition on factors set $\mathcal{S}$. We take Shapes3D as an example when DisDiff sampling images conditioned on the background color is red. We obtain images of the background color red and other factors randomly sampled. Figure 4 shows that DisDiff can condition individual factors on Shapes3D. In addition, DisDiff also has such ability on the real-world dataset (CelebA) in Figure 4. DisDiff is capable of sampling information exclusively to conditions.

# 6    Limitations

Our method is completely unsupervised, and without any guidance, the learned disentangled representations on natural image sets may not be easily interpretable by humans. We can leverage models like CLIP as guidance to improve interpretability. Since DisDiff is a diffusion-based method, compared to the VAE-based and GAN-based, the generation speed is slow, which is also a common problem for DPM-based methods. The potential negative societal impacts are malicious uses.

# 7    Conclusion

In this paper, we demonstrate a new task: disentanglement of DPM. By disentangling a DPM into several disentangled gradient fields, we can improve the interpretability of DPM. To solve the task, we build an unsupervised diffusion-based disentanglement framework named DisDiff. DisDiff learns a disentangled representation of the input image in the diffusion process. In addition, DisDiff learns a disentangled gradient field for each factor, which brings the following new properties for disentanglement literature. DisDiff adopted disentangling constraints on all different timesteps, a new inductive bias. Except for image editing, with the disentangled DPM, we can also sample partial conditions on the part of the information by superpositioning the sub-gradient field. Applying DisDiff to more general conditioned DPM is a direction worth exploring for future work. Besides, utilizing the proposed disentangling method to pre-trained conditional DPM makes it more flexible.

## Acknowledgement

We thank all the anonymous reviewers for their constructive and helpful comments, which have significantly improved the quality of the paper. The work was partly supported by the National Natural Science Foundation of China (Grant No. 62088102).

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

# Appendix

## A    The Training and Sampling Algorithms

As shown in Algorithm 1, we also present the training procedure of DisDiff. In addition, Algorithms 2 and 3 show the sampling process of DisDiff in the manner of DDPM and DDIM under the condition of $S$.

---
**Algorithm 1** Training procedure
---
**Prepare:** dataset distribution $p(x_0)$, pre-trained DPM $\epsilon_\theta$, pre-trained VQ-VAE, $E_{VQ}, D_{VQ}$,
**Initialize:** encoder $E_\phi$, decoder $G_\psi$.
**repeat**
    $x_0 \sim p(x_0)$
    $z_{x0} = E_{VQ}(x_0)$
    $t \sim \text{Uniform}(0, 1, \dots, T)$
    $\epsilon \sim \mathcal{N}(0, I)$
    $z_{xt} = \sqrt{\bar{\alpha}_t} z_{x0} + \sqrt{1 - \bar{\alpha}_t}\epsilon$
    $c \sim \text{Uniform}(1, 2, \dots, N)$
    $z = E_\phi(x_0)$
    $\mathcal{L}_r = \|\epsilon - \epsilon_\theta(z_{xt}, t) + \frac{\sqrt{\alpha_t}\sqrt{1-\bar{\alpha}_t}}{\beta_t}\sigma_t \sum_{i \in \mathcal{C}} G_\psi(z_{xt}, z^i, t, i)\|$

    Use $z_{xt}, t, E_\phi, E_{VQ}, D_{VQ}$ and $c$ to calculate $\mathcal{L}_{in}$ and $\mathcal{L}_{va}$ by Eq. 12 and Eq. 14.
    Calculate weight $\gamma_d$ by Eq. 17
    Update by taking gradient descent step on
    $\nabla_{\phi,\psi}(\mathcal{L}_r + \gamma_d(\mathcal{L}_{in} + \mathcal{L}_{va}))$
**until** $converged$;

---

---
**Algorithm 2** DDPM Sampling
---
**Prepare:** pre-trained DPM $\epsilon_\theta$, pre-trained VQ-VAE, $E_{VQ}, D_{VQ}$, the sampling factor set $S \subseteq C$,
**Input:** image $x$
$z = E_\phi(x)$
$z_{xT} \sim \mathcal{N}(0, I)$
**for** $t = T$ **to** 1 **do**
  **if** $t \neq 1$ **then**
    $\epsilon \sim \mathcal{N}(0, I)$
  **else**
    $\epsilon = 0$
  **end if**
  $z_{x(t-1)} = \frac{1}{\sqrt{\alpha_t}}\left[z_{xt} - \frac{\beta_t}{\sqrt{1-\alpha_t}}\epsilon_\theta(z_{xt}, t)\right] + \sigma_t^2 \sum_{i \in S} G_\phi(z_{xt}, z^i, t, i) + \sigma_t\epsilon$
**end for**
**return** $D_{VQ}(z_{x0})$

---

## B    Training Details

**Encoder** We use the encoder architecture following DisCo [36] for a fair comparison. The details of the encoder are presented in Table 3, which is identical to the encoder of $\beta$-TCVAE and FactorVAE in Table 1.

**Decoder** We use the decoder architecture following PADE [49]. The details of the decoder are presented in Table 4, which is identical to the Unet decoder in the pre-trained DPM. **DPM** We adopted the VQ-version latent diffusion for image resolution of 64 and 256 as our pre-trained DPM. The details of the DPM are presented in Table 4. During the training of DisDiff, we set the batch size as 64 for all datasets. We always set the learning rate as $1e-4$. We use EMA on all model

---

**Algorithm 3** DDIM Sampling

---

**Prepare:** pre-trained DPM $\epsilon_\theta$, pre-trained VQ-VAE $E_{VQ}, D_{VQ}$, the sampling factor set $S \subseteq C$,
**Input:** sample $x$, sampling sequence $\{t_i\}_{i=1}^K$, and $t_1 = 0, t_K = T$
$z = E_\phi(x)$
$z_{xT} \sim \mathcal{N}(0, I)$ or inferred $z_{xT}$ of inverse DDIM
**for** $j = K$ **to** $2$ **do**
$\quad \epsilon_\theta(z_{xt_j}, z^S, t_j) = \epsilon_\theta(z_{xt_j}, t) - \sum_{i \in S} \sqrt{1 - \bar{\alpha}_{t_j}} G_\psi(z_{xt_j}, z^i, t_j, i)$
$\quad z_{xt_{j-1}} = \sqrt{\bar{\alpha}_{t_{j-1}}} \left( \frac{z_{xt_j} - \sqrt{1 - \bar{\alpha}_{t_j}} \epsilon_\theta(z_{xt_j}, z^S, t_j)}{\sqrt{\alpha_{t_j}}} \right) + \sqrt{1 - \bar{\alpha}_{t_{j-1}}} \epsilon_\theta(z_{xt_j}, z^S, t_j)$
**end for**
**return** $D_{VQ}(z_{x0})$

---

parameters with a decay factor of 0.9999. Following DisCo [36], we set the representation vector dimension to 32.

Table 4: Encoder E architecture used in DisDiff.

| |
|---|
| Conv $7 \times 7 \times 3 \times 64$, stride $= 1$ |
| ReLu |
| Conv $4 \times 4 \times 64 \times 128$, stride $= 2$ |
| ReLu |
| Conv $4 \times 4 \times 128 \times 256$, stride $= 2$ |
| ReLu |
| Conv $4 \times 4 \times 256 \times 256$, stride $= 2$ |
| ReLu |
| Conv $4 \times 4 \times 256 \times 256$, stride $= 2$ |
| ReLu |
| FC $4096 \times 256$ |
| ReLu |
| FC $256 \times 256$ |
| ReLu |
| FC $256 \times K$ |

Table 5: Decoder (pre-trained DPM) architecture used in DisDiff.

| Parameters | Shapes3D / Cars3D/ MPI3D | FFHQ | CelebA-HQ |
|---|---|---|---|
| Base channels | 16 | 64 | 64 |
| Channel multipliers | [ 1,2,4,4 ] | [1,2,3,4] | [1,2,3,4] |
| Attention resolutions | [ 1, 2, 4] | [2,4,8] | [2,4,8] |
| Attention heads num | 8 | 8 | 8 |
| Model channels | 64 | 224 | 224 |
| Dropout | 0.1 | | |
| Images trained | 0.48M/0.28M/1.03M | 130M | 52M |
| $\beta$ scheduler | Linear | | |
| Training T | 1000 | | |
| Diffusion loss | MSE with noise prediction $\epsilon$ | | |

# C  Visualizations on Cars3D.

We also provide the qualitative results on Cars3D in Figure 5. We can observe that on the artificial dataset Cars3D, DisDiff also learns disentangled representation. In addition, we also provide the results of swapping representations that contain no information.

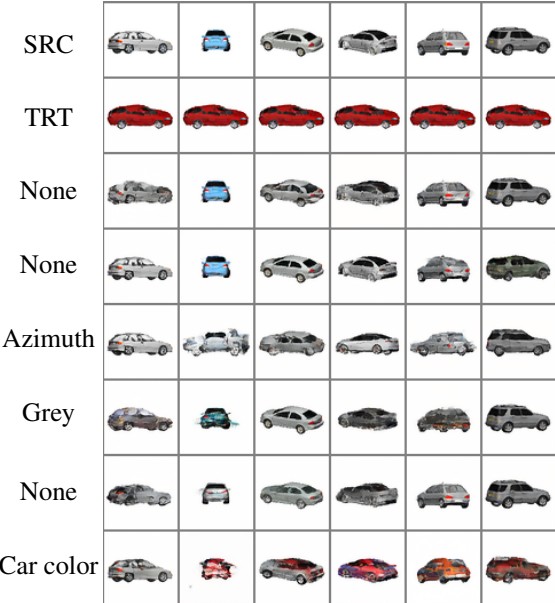

Figure 5: The qualitative results on Cars3D. The source (SRC) images provide the representations of the generated image. The target (TRT) image provides the representations for swapping. The remaining rows of images are generated by swapping the representation of the corresponding factor on Shapes3D. DisDiff learns pure factors by each representation. The learned factors are None, None, azimuth, grey, and car color.

## D    Visualizations on MPI3D.

We provide the qualitative results on MPI3D in Figure 6. We can observe that even on such a challenging dataset (MPI3D) of disentanglement, DisDiff also learns disentangled representation. In addition, we also provide three rows of images swapping representations that contain no information.

## E    Orthogonal-constrained DisDiff

We follow DisCo [36] and LD [43] to construct learnable orthogonal directions of the bottleneck layer of Unet. The difference is that we use QR decomposition rather than the orthonormal matrix proposed by LD. Specifically, for a learnable matrix $A \in \mathbb{R}^{C \times d}$, where $d$ is the dimension of direction, we have the following equation:

$$A = QR \tag{19}$$

where $Q \in \mathbb{R}^{C \times d}$ is an orthogonal matrix, and $R \in \mathbb{R}^{d \times d}$ is an upper triangular matrix, which is discarded in our method. We use $Q$ to provide directions in Figure 7. Since the directions are sampled, and we follow DisCo and LD to discard the image reconstruction loss, we use the DDIM inverse ODE to obtain the inferred $x_T$ for image editing. The sampling process is shown in Algorithm 3.

## F    Qualitative Results on CelebA-HQ

In order to demonstrate the effectiveness of Orthogonal-constrained DisDiff, we train it on dataset CelebA-HQ. We also conduct our experiments with the latent unit number of 10. The results are shown in Figure 9. Since the real-world datasets have many factors that are more than 10, we also conduct our experiment with the latent unit number of 30. As shown in Figure 11, Orthogonal-constrained DisDiff learns consistent factors for different images. There are more factors than Figure 9 shows. Therefore, we conduct another experiment using another random seed with a latent unit number of 10. The results are shown in Figure 10.

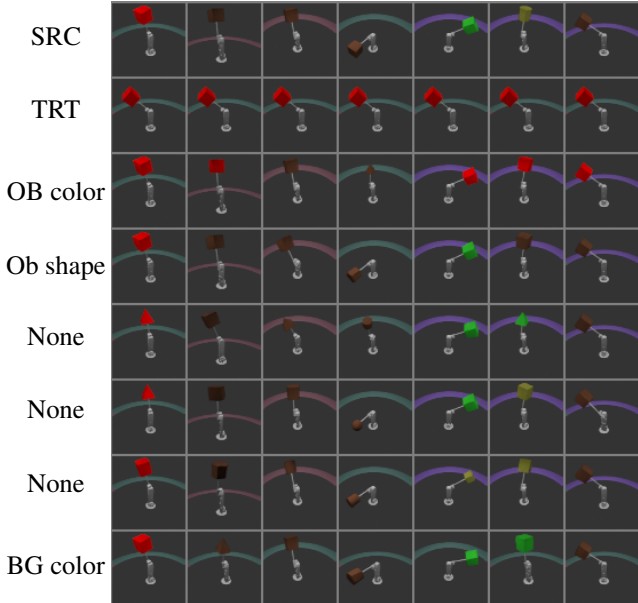

Figure 6: The qualitative results on MPI3D. The source (SRC) images provide the representations of the generated image. The target image provides the representations for swapping. The remaining rows of images are generated by swapping the representations of the corresponding factor on MPI3D. DisDiff learns pure factors by each representation. The learned factors are object color (OB color), object shape (OB shape), None, None, None, and background color (BG color).

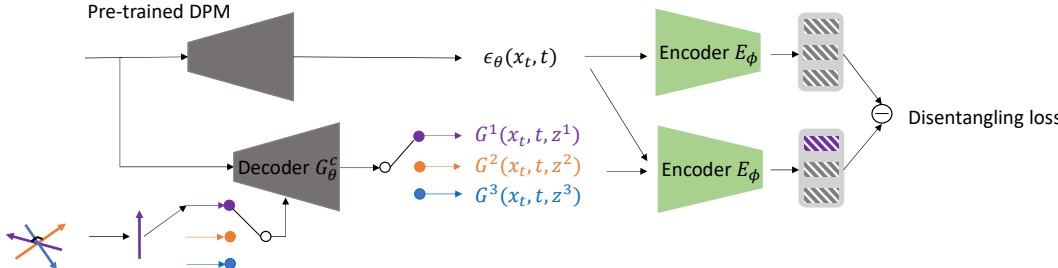

Figure 7: Illustration of Orthogonal-constrained DisDiff. In this architecture, we use a powerful decoder to replace the original decoder in DisDiff. Since we train the decoder from scratch, the ability to disentanglement on large natural datasets is limited. However, we can use the pre-trained Unet decoder as our powerful decoder by feeding a set of learnable orthogonal directions to make it output the corresponding gradient field. These directions and the decoder compose a powerful decoder that takes directions as input and generates the corresponding gradient fields as output.

## G  CLIP-guided DisDiff

To tackle the disentanglement in real-world settings, we propose replacing the encoder in DisDiff with a powerful encoder, which lets us leverage the knowledge of the pre-trained model to facilitate the disentanglement. Specifically, we adopt CLIP as the encoder. Since the CLIP is an image-text pair pre-trained model, the multi-modal information is the pre-trained information we can leverage. Therefore, we use the CLIP text encoder to provide the condition but use the CLIP image encoder to calculate the disentangling loss. Figure 8 shows the obtained new architecture.

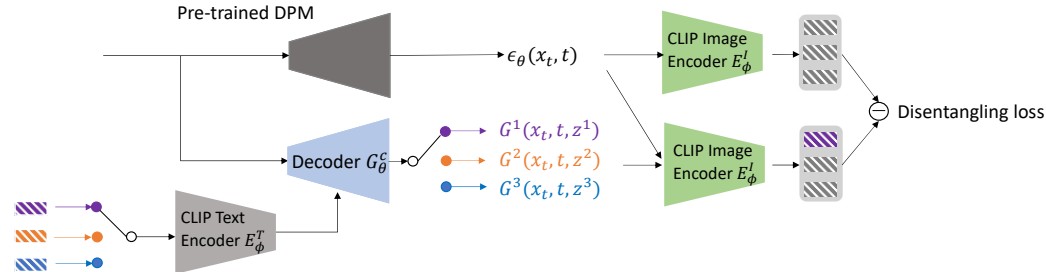

Figure 8: Illustration of CLIP-guided DisDiff. In this architecture, a powerful encoder replaces the original one in DisDiff. Specifically, we adopt the CLIP Image encoder as our new encoder, which also serves as guidance for ASYRP [25]. By leveraging knowledge learned from pre-trained CLIP, DisDiff can learn meaningful gradient fields. To use the learned knowledge of the language-image pair contrastive pre-training, we utilize attribute words as inputs, where different words correspond to different gradient fields. Finally, we incorporate disentangling loss at the end of the encoder to enforce both the invariant and variant constraints.

## H  Qualitative Results on FFHQ

We train the CLIP-guided DisDiff on FFHQ to verify its effectiveness. Since the attribute words used in the training of the CLIP-guided DisDiff are unpaired and provided in ASYRP, the learning of these factors is still unsupervised. As discussed in Section 4.2, DisDiff can do partial sampling that is conditioned on the part of the factors. Similarly, as shown in Figure 13, CLIP-guided DisDiff also has such an ability. In addition, as shown in Figure 12, we can also use the trained decoder of CLIP-guided DisDiff to edit the images by using the inverse ODE of DDIM to infer the $x_T$.

## I  The Latent Number < The Number of GT Factors

When the latent number is smaller than the number of factors, the model's performance will significantly decrease, which is in accordance with our intuition. However, the performance of other methods will also decrease under this condition. We conducted experiments using FactorVAE and $\beta$-TCVAE to demonstrate it. We trained our model, FactorVAE and $\beta$-TCVAE on Shapes3D (The number of factors is 6) with a latent number of 3, and the hyperparameters and settings in their original paper were adopted for the latter two models accordingly. As shown in Table 6, our model showed a lower degree of disentanglement reduction than these two models and was significantly lower than $\beta$-TCVAE.

Table 6: The disentanglement performance when the latent number is 3 (mean $\pm$ std, higher is better).

| Model | DCI score | FactorVAE score |
|---|---|---|
| $\beta$-TCVAE | $0.098 \pm 0.043$ | $0.383 \pm 0.0444$ |
| FactorVAE | $0.3113 \pm 0.014$ | $0.4982 \pm 0.0049$ |
| DisDiff (ours) | $\mathbf{0.3279 \pm 0.023}$ | $\mathbf{0.4986 \pm 0.0054}$ |

## J  Quantitative Results on More Metrics

We experiment with MIG and MED metrics on the Shapes3D dataset, two classifier-free metrics. For MIG, we utilized the implementation provided in `disentanglement_lib`[4], while for MED, we used the official implementation available at Github[5]. The experimental results in Table 7 indicate that our proposed method still has significant advantages, especially in MED-topk and MIG.

---

[4]https://github.com/google-research/disentanglement_lib

[5]https://github.com/noahcao/disentanglement_lib_med

Table 7: Comparisons of disentanglement on the additional metrics: MED, MED-topk, and MIG (mean $\pm$ std, higher is better). Except for DisCo and DisDiff, other results (MED and MED-topk) are taken from the MED [4]. The missing value is due to the need for publicly released pre-trained models.

| Model | MED | MED-topk | MIG |
|---|---|---|---|
| $\beta$-VAE | $52.5 \pm 9.4$ | $19.2 \pm 1.4$ | – |
| $\beta$-TCVAE | $53.2 \pm 4.9$ | $25.0 \pm 0.5$ | $0.406 \pm 0.175$ |
| FactorVAE | $55.9 \pm 8.0$ | $8.2 \pm 4.0$ | $0.434 \pm 0.143$ |
| DIP-VAE-I | $43.5 \pm 3.7$ | $16.2 \pm 0.9$ | – |
| DIP-VAE-II | $52.6 \pm 5.2$ | – | – |
| AnnealedVAE | $56.1 \pm 1.5$ | – | – |
| MoCo | – | $18.1 \pm 0.6$ | – |
| MoCov2 | – | $13.6 \pm 1.7$ | – |
| BarlowTwins | – | $20.0 \pm 0.3$ | – |
| SimSiam | – | $30.0 \pm 2.0$ | – |
| BYOL | – | $19.7 \pm 1.3$ | – |
| InfoGAN-CR | – | – | $0.297 \pm 0.124$ |
| LD | – | – | $0.168 \pm 0.056$ |
| DS | – | – | $0.356 \pm 0.090$ |
| DisCo | $38.0 \pm 2.7$ | $54.8 \pm 2.1$ | $0.512 \pm 0.068$ |
| DisDiff (Ours) | $\mathbf{59.5 \pm 1.1}$ | $\mathbf{64.5 \pm 1.6}$ | $\mathbf{0.5671 \pm 0.033}$ |

# K    Proof of Proposition 1

**Background**: Our encoder transforms the data $x_0$ (hereafter denoted as $x$, the encoder keeps fixed for disentangling loss) into two distinct representations $z^c$ and $z^k$. We assume that the encoding distribution follows a Gaussian distribution $p(z^c|x)$, with a mean of $E^c(x)$. However, the distribution $p(x|z^c)$ is intractable. We can employ conditional sampling of our DPM $q_\theta(x|z^c)$ to approximate the sampling from the distribution $p(x|z^c)$. Given that we optimize the disentangling loss at each training step, we implement Eq. 15 and Eq. 16 in the main paper as an efficient alternative to sample from $q_\theta(x|z^c)$. We further assume that the distribution used to approximate $p(z^k|z^c)$ follows a Gaussian distribution $q_\theta(z^k|z^c)$, with a mean of $z^{k|c} = E^k(D(z^c))$, where $D$ represents the operations outlined in Eq. 15 and Eq. 16. We represent the pretrained unconditional DPM using the notation $q_\theta(x) \approx p(x)$.

**Proposition 2.** *For a PGM: $z^c \rightarrow x \rightarrow z^k$, we can minimize the upper bound (estimator) for the mutual information between $z^c$ and $z^k$ (where $k \neq c$) by the following objective:*

$$\min_{k,c,x_0,\hat{x}_0^c} \mathbb{E} \, \|\hat{z}^{k|c} - \hat{z}^k\| - \|\hat{z}^{k|c} - z^k\| + \|\hat{z}^{c|c} - z^c\| \tag{20}$$

Considering the given context, we can present the PGM related to $q$ as follows: $z^c \rightarrow x \rightarrow z^k$. The first transition is implemented by DPM represented by the distribution $q_\theta$. Subsequently, the second transition is implemented by the encoder, denoted by the distribution $p$.

**Proposition 1.1** The estimator below is an upper bound of the mutual information $I(z^c, z^k)$. For clarity, we use expectation notation with explicit distribution.

$$I_{est} = \mathbb{E}_{p(z^c,z^k)}[\log p(z^k|z^c)] - \mathbb{E}_{p(z^c)p(z^k)}[\log p(z^k|z^c)]. \tag{21}$$

Equality is attained if and only if $z^k$ and $z^c$ are independent.

*Proof*: $I_{est} - I(z^c, z^k)$ can be formulated as follows:

$$\mathbb{E}_{p(z^c,z^k)}[\log p(z^k|z^c)] - \mathbb{E}_{p(z^c)p(z^k)}[\log p(z^k|z^c)] - \mathbb{E}_{p(z^c,z^k)}[\log p(z^k|z^c) - \log p(z^k)]$$
$$= \mathbb{E}_{p(z^c,z^k)}[\log p(z^k)] - \mathbb{E}_{p(z^c)p(z^k)}[\log p(z^k|z^c)] \tag{22}$$
$$= \mathbb{E}_{p(z^k)}[\log p(z^k) - \mathbb{E}_{p(z^c)}[\log p(z^k|z^c)]].$$

Using Jensen's Inequality, we obtain $\log p(z^k) = \log(\mathbb{E}_{p(z^c)}[p(z^k|z^c)]) \geqslant \mathbb{E}_{p(z^c)}[\log p(z^k|z^c)]$. Therefore, the following holds:

$$I_{est} - I(z^c, z^k) = \mathbb{E}_{p(z^k)}[\log p(z^k) - \mathbb{E}_{p(z^c)}[\log p(z^k|z^c)]] \geqslant 0. \qquad (23)$$

Since the distribution $p(z^k|z^c)$ is intractable, we employ $q_\theta(z^k|z^c)$ to approximate $p(z^k|z^c)$.

$$I_\theta = \mathbb{E}_{p(z^c,z^k)}[\log q_\theta(z^k|z^c)] - \mathbb{E}_{p(z^c)p(z^k)}[\log q_\theta(z^k|z^c)]. \qquad (24)$$

**Proposition 1.2**: The new estimator can be expressed as follows:

$$I_\theta = -\mathbb{E}_{p(z^c,z^k)}\|z^k - \hat{z}^{k|c}\| + \mathbb{E}_{p(z^c)}\mathbb{E}_{q_\theta(\hat{x})}\mathbb{E}_{p(\hat{z}^k|\hat{x})}\|\hat{z}^k - \hat{z}^{k|c}\| + C_1. \qquad (25)$$

The first term corresponds to $d_k^p$ in Eq. 13 of the main paper, while the second term corresponds to Eq. 11 in the main paper.

*Proof*:

Considering the first term, we obtain:

$$\log q_\theta(z^k|z^c) = \log \exp(-\|z^k - E^k(D(z^c))\|) + C_2 = -\|z^k - \hat{z}^{k|c}\| + C_2. \qquad (26)$$

For the second term, we obtain:

$$\begin{aligned}
\mathbb{E}_{p(z^k)}[\log q_\theta(z^k|z^c)] &= \int \int p(x, z^k) \log q_\theta(z^k|z^c) dz^k dx \\
&= \int p(x) \int p(z^k|x) \log q_\theta(z^k|z^c) dz^k dx \\
&\approx \int q_\theta(\hat{x}) \int p(\hat{z}^k|\hat{x}) \log q_\theta(\hat{z}^k|z^c) d\hat{z}^k d\hat{x} \qquad (27) \\
&= \mathbb{E}_{q_\theta(\hat{x})}\mathbb{E}_{p(\hat{z}^k|\hat{x})}[\log q_\theta(\hat{z}^k|z^c)] \\
&= \mathbb{E}_{q_\theta(\hat{x})}\mathbb{E}_{p(\hat{z}^k|\hat{x})}\|\hat{z}^k - E^k(D(z^c))\| + C_3 \\
&= \mathbb{E}_{q_\theta(\hat{x})}\mathbb{E}_{p(\hat{z}^k|\hat{x})}\|\hat{z}^k - \hat{z}^{k|c}\| + C_3.
\end{aligned}$$

**Proposition 1.3**: If the following condition is fulfilled, the new estimator $I_\theta$ can serve an upper bound for the mutual information $I(z^c, z^k)$, i.e., $I_\theta \geqslant I(z^c, z^k)$. The equality holds when $z^k$ and $z^c$ are independent. We denote $q_\theta(z^k|z^c)p(z^c)$ as $q_\theta(z^k, z^c)$.

$$D_{KL}(p(z^k, z^c)\|q_\theta(z^k, z^c)) \leqslant D_{KL}(p(z^k)p(z^c)\|q_\theta(z^k, z^c)). \qquad (28)$$

*Proof*:

$I_\theta - I(z^c, z^k)$ can be formulated as follows:

$$\begin{aligned}
&\mathbb{E}_{p(z^c,z^k)}[\log q_\theta(z^k|z^c)] - \mathbb{E}_{p(z^c)p(z^k)}[\log q_\theta(z^k|z^c)] - \mathbb{E}_{p(z^c,z^k)}[\log p(z^k|z^c) - \log p(z^k)] \\
&= \mathbb{E}_{p(z^c,z^k)}[\log q_\theta(z^k|z^c) - \log p(z^k|z^c)] + \mathbb{E}_{p(z^c)p(z^k)}[-\log q_\theta(z^k|z^c) + \log p(z^k)] \\
&= \mathbb{E}_{p(z^c)p(z^k)}[-\log q_\theta(z^k|z^c) + \log p(z^k)] - \mathbb{E}_{p(z^c,z^k)}[\log p(z^k|z^c) - \log q_\theta(z^k|z^c)] \qquad (29) \\
&= \mathbb{E}_{p(z^c)p(z^k)}\left[\log \frac{p(z^k)p(z^c)}{q_\theta(z^k|z^c)p(z^c)}\right] - \mathbb{E}_{p(z^c,z^k)}\left[\log \frac{p(z^k|z^c)p(z^c)}{q_\theta(z^k|z^c)p(z^c)}\right] \\
&= D_{KL}(p(z^c)p(z^k)\|q_\theta(z^k, z^c)) - D_{KL}(p(z^c, z^k)\|q_\theta(z^k, z^c)).
\end{aligned}$$

If $D_{KL}(p(z^k, z^c)\|q_\theta(z^k, z^c)) \leqslant D_{KL}(p(z^k)p(z^c)\|q_\theta(z^k, z^c))$, we obtain:

$$I_\theta \geqslant I(z^c, z^k). \qquad (30)$$

Since the condition is not yet applicable, we will continue exploring alternative conditions.

**Proposition 1.4**: If $D_{KL}(p(z^k|z^c)\|q_\theta(z^k|z^c)) \leqslant \epsilon$, then, either of the following conditions holds: $I_\theta \geqslant I(z^c, z^k)$ or $|I_\theta - I(z^c, z^k)| \leqslant \epsilon$.

*Proof*: If $D_{KL}(p(z^k, z^c)||q_\theta(z^k, z^c)) \leqslant D_{KL}(p(z^k)p(z^c)||q_\theta(z^k, z^c))$, then

$$I_\theta \geqslant I(z^c, z^k), \tag{31}$$

else:

$$
\begin{aligned}
D_{KL}(p(z^k)p(z^c)||q_\theta(z^k, z^c)) &< D_{KL}(p(z^k, z^c)||q_\theta(z^k, z^c)) \\
&= \mathbb{E}_{p(z^c, z^k)} \left[ \log \frac{p(z^k|z^c)}{q_\theta(z^k|z^c)} \right] \\
&= D_{KL}(p(z^k|z^c)||q_\theta(z^k|z^c)) \\
&\leqslant \epsilon.
\end{aligned}
\tag{32}
$$

$$|I_\theta - I(z^c, z^k)| < \max(D_{KL}(p(z^k, z^c)||q_\theta(z^k, z^c)), D_{KL}(p(z^k)p(z^c)||q_\theta(z^k, z^c))) \leqslant \epsilon. \tag{33}$$

**Proposition 1.5**: Consider a PGM $z^c \to x \to z^k$; if $D_{KL}(q_\theta(x|z^c)|p(x|z^c)) = 0$, it follows that $D_{KL}(p(z^k|z^c)||q_\theta(z^k|z^c)) = 0$.

*Proof*: if $D_{KL}(q_\theta(x|z^c)|p(x|z^c)) = 0$, we have $q_\theta(x|z^c) = p(x|z^c)$.

$$
\begin{aligned}
D_{KL}(p(z^k|z^c)||q_\theta(z^k|z^c)) &= \mathbb{E}_{p(z^c, z^k)} \left[ \log \frac{p(z^k|z^c)}{q_\theta(z^k|z^c)} \right] \\
&= \mathbb{E}_{p(z^c, z^k)} \left[ \log \frac{\int p(x|z^c)p(z^k|x)dx}{\int q_\theta(x|z^c)p(z^k|x)dx} \right] \\
&= \mathbb{E}_{p(z^c, z^k)}[\log 1] = 0.
\end{aligned}
\tag{34}
$$

Ideally, by minimizing $D_{KL}(q_\theta(x|z^c)|p(x|z^c))$ to 0, we can establish $I_\theta$ either as an upper bound or as an estimator of $I(z^c, z^k)$.

**Proposition 1.6**: we can minimize $D_{KL}(q_\theta(x|z^c)|p(x|z^c))$ by minimizing the following objective:

$$\mathbb{E}_{p(z^c)}\mathbb{E}_{q_\theta(x|z^c)}\|z^c - \hat{z}^{c|c}\| + C_4. \tag{35}$$

Note that the equation above exactly corresponds to $d_c^p$ in Eq. 13, and $d_k^n$ is a constant.

*Proof*:

$$D_{KL}(q_\theta(x|z^c)|p(x|z^c)) = \int p(z^c) \int q_\theta(x|z^c) \log \frac{q_\theta(x|z^c)}{p(x|z^c)} dx dz^c. \tag{36}$$

We focus on the following term:

$$\int q_\theta(x|z^c) \log \frac{q_\theta(x|z^c)}{p(x|z^c)} dx = \int q_\theta(x|z^c)[\log q_\theta(x|z^c) - \log p(x|z^c)]dx. \tag{37}$$

Applying the Bayes rule, we obtain:

$$
\begin{aligned}
&= \int q_\theta(x|z^c)[\log q_\theta(x|z^c) - \log \frac{p(z^c|x)p(x)}{p(z^c)}]dx \\
&= \int q_\theta(x|z^c) \left[ \log \frac{q_\theta(x|z^c)}{p(x)} - \log p(z^c|x) + \log p(z^c) \right] dx \\
&= D_{KL}(q_\theta(x|z^c)|p(x)) - \mathbb{E}_{q_\theta(x|z^c)}[\log p(z^c|x)] + \log p(z^c) \\
&\leqslant -\mathbb{E}_{q_\theta(x|z^c)}[\log p(z^c|x)] + \log p(z^c) \\
&= -\mathbb{E}_{q_\theta(x|z^c)}[\log \exp(-|z^c - E^c(x)|)] + \log p(z^c) + C_5 \\
&= \mathbb{E}_{q_\theta(x|z^c)}\|z^c - \hat{z}^{c|c}\| + \log p(z^c) + C_5.
\end{aligned}
\tag{38}
$$

We can minimize $\mathbb{E}_{p(z^c)}\mathbb{E}_{q_\theta(x|z^c)}\|z^c - \hat{z}^{c|c}\|$ instead.

Note that the loss usually cannot be optimized to a global minimum in a practical setting, but how the gap influences $I_\theta$ is beyond the scope of this paper. We leave it as future work. Figure 4 in the main paper shows that our model has successfully fitted the distribution of $p(x|z^c)$ using $q_\theta(x|z^c)$. Therefore, we can understand that the model's loss can be regarded as $I_{est}$.

Table 8: Ablation study of DisDiff on image encoder, components, batchsize and token numbers on Shapes3D.

| Method | SSIM(↑) | LPIPS(↓) | MSE(↓) | DCI(↑) | Factor VAE(↑) |
|---|---|---|---|---|---|
| PDAE | 0.9830 | 0.0033 | 0.0005 | 0.3702 | 0.6514 |
| DiffAE | **0.9898** | 0.0015 | $\mathbf{8.1983e-05}$ | 0.0653 | 0.1744 |
| DisDiff (ours) | 0.9484 | **0.0006** | $9.9293e-05$ | **0.723** | **0.902** |

## L  Comparison with Diffusion-based Autoencoders

We include a comparison with more recent diffusion baselines, such as PADE [49] and DiffAE [33]. The results are shown in Table 8.

The relatively poor disentanglement performance of the two baselines can be attributed to the fact that neither of these models was specifically designed for disentangled representation learning, and thus, they generally do not possess such capabilities. DiffAE primarily aims to transform the diffusion model into an autoencoder structure, while PDAE seeks to learn an autoencoder for a pre-trained diffusion model. Unsupervised disentangled representation learning is beyond their scope.

Traditional disentangled representation learning faces a trade-off between generation quality and disentanglement ability (e.g., betaVAE [16], betaTCVAE [5], FactorVAE [23]). However, fortunately, disentanglement in our model does not lead to a significant drop in generation quality; in fact, some metrics even show improvement. The reason for this, as described in DisCo [36], is that using a well-pre-trained generative model helps avoid this trade-off.

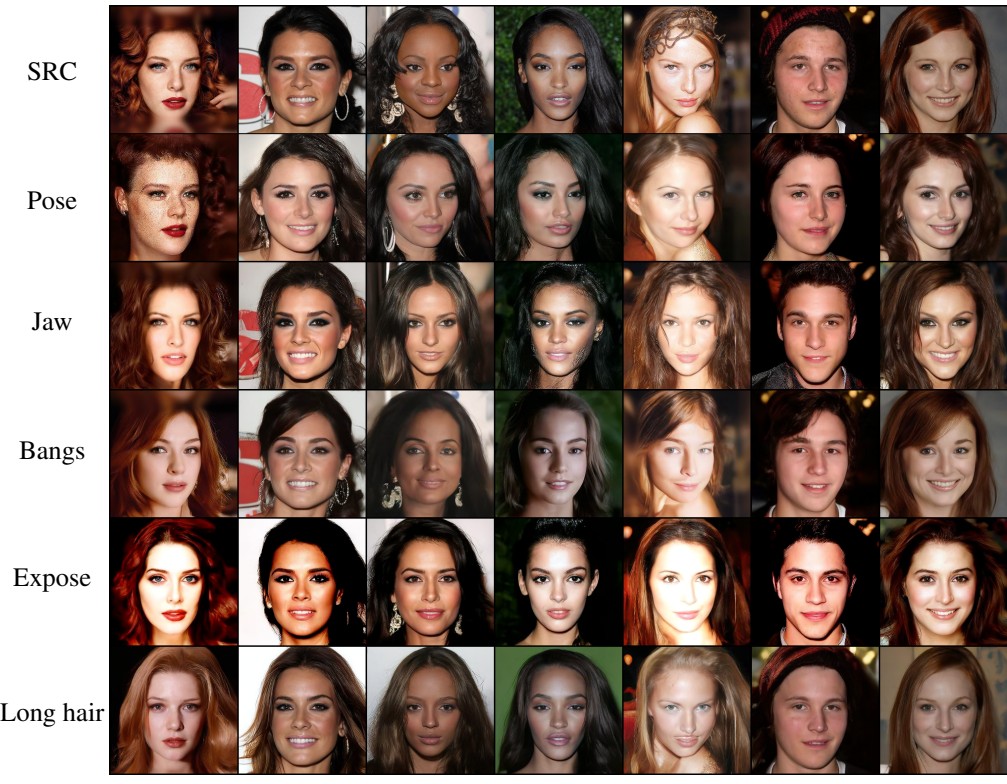

Figure 9: The qualitative results of orthogonal constraint on CelebA-HQ (seed 0). The source (SRC) row provides the images of the generated image. The remaining rows of images are generated by shifting the representation along the learned direction of the corresponding factor on CelebA-HQ. DisDiff learns the direction of pure factors unsupervised. The learned factors are listed in the leftmost column. The images are sampled by using the inferred $x_T$.

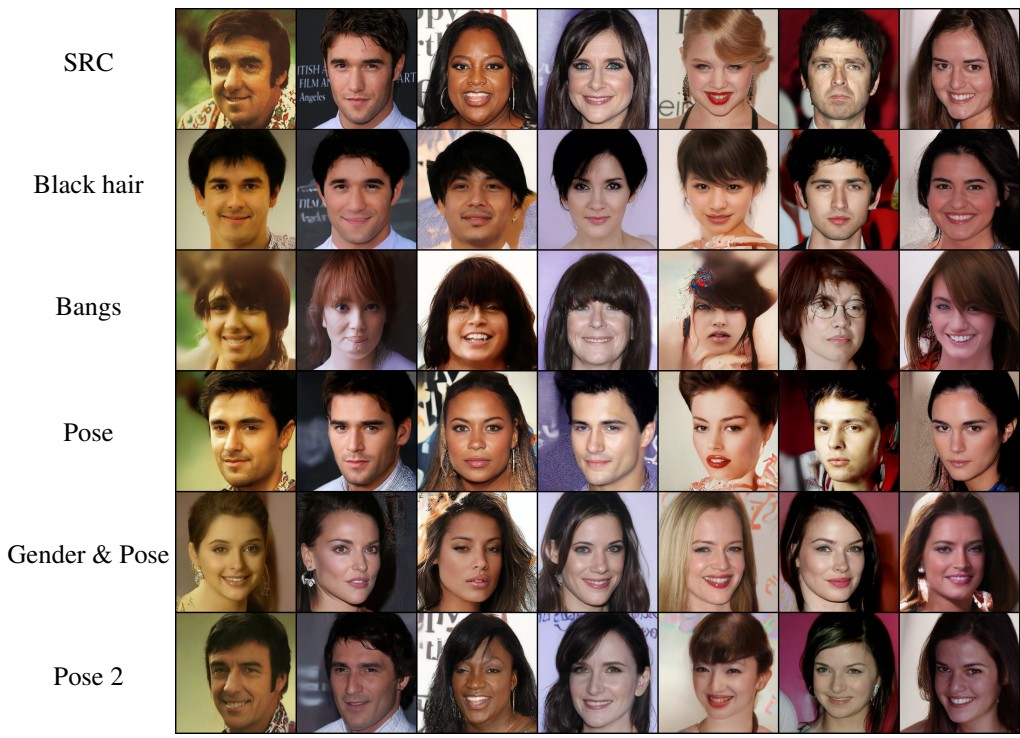

Figure 10: The qualitative results of orthogonal constraint on CelebA-HQ (seed 1). The source (SRC) row provides the images of the generated image. The remaining rows of images are generated by shifting the representation along the learned direction of the corresponding factor on CelebA-HQ. DisDiff learns the direction of pure factors unsupervised. The learned factors are listed in the leftmost column. The images are sampled by using the inferred $x_T$.

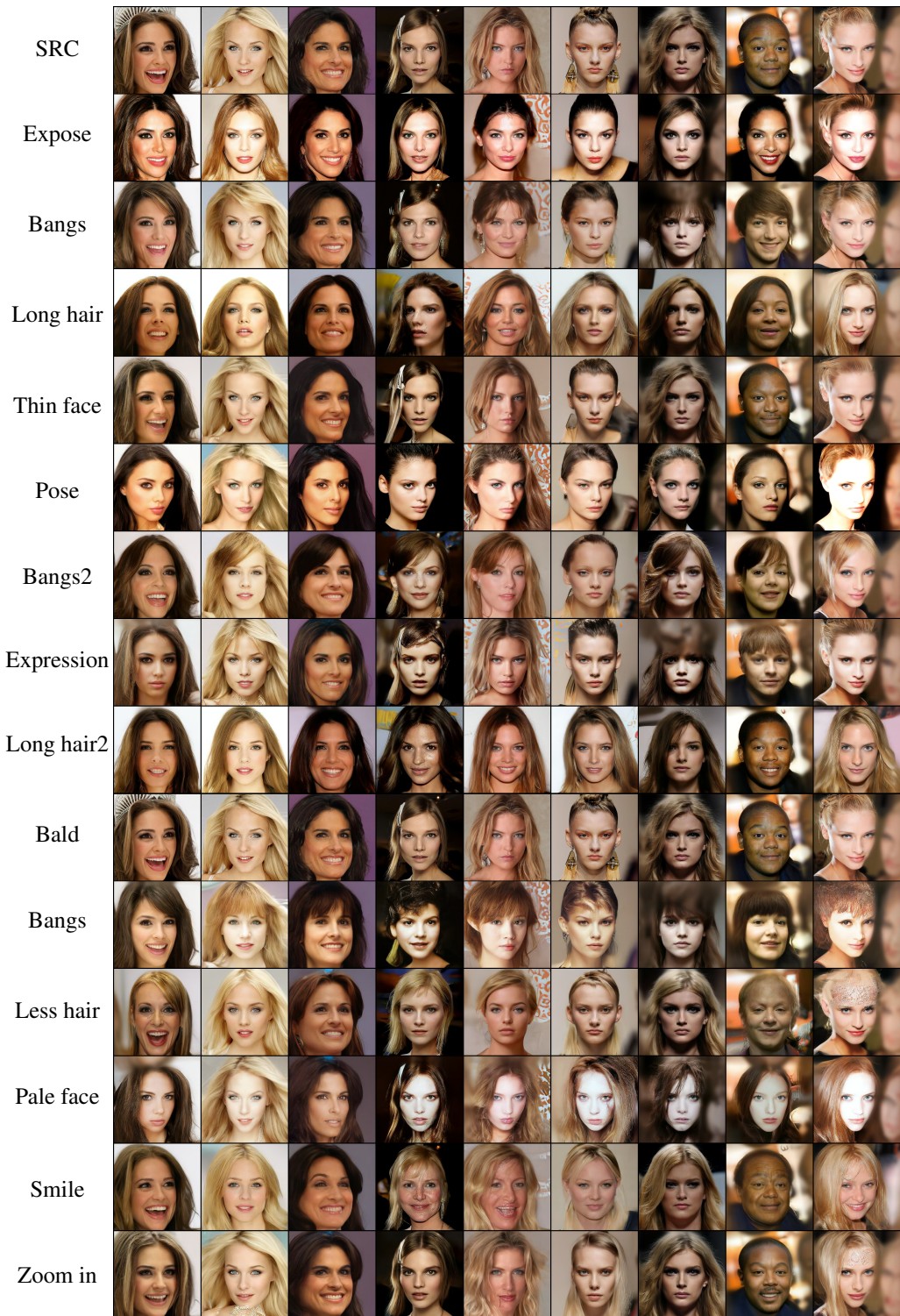

Figure 11: The qualitative results of orthogonal constraint on CelebA-HQ (latent number = 30). The source (SRC) row provides the images of the generated image. The remaining rows of images are generated by shifting the representation along the learned direction of the corresponding factor on CelebA-HQ. DisDiff learns the direction of pure factors unsupervised. The learned factors are listed in the leftmost column. The images are sampled by using the inferred $x_T$.

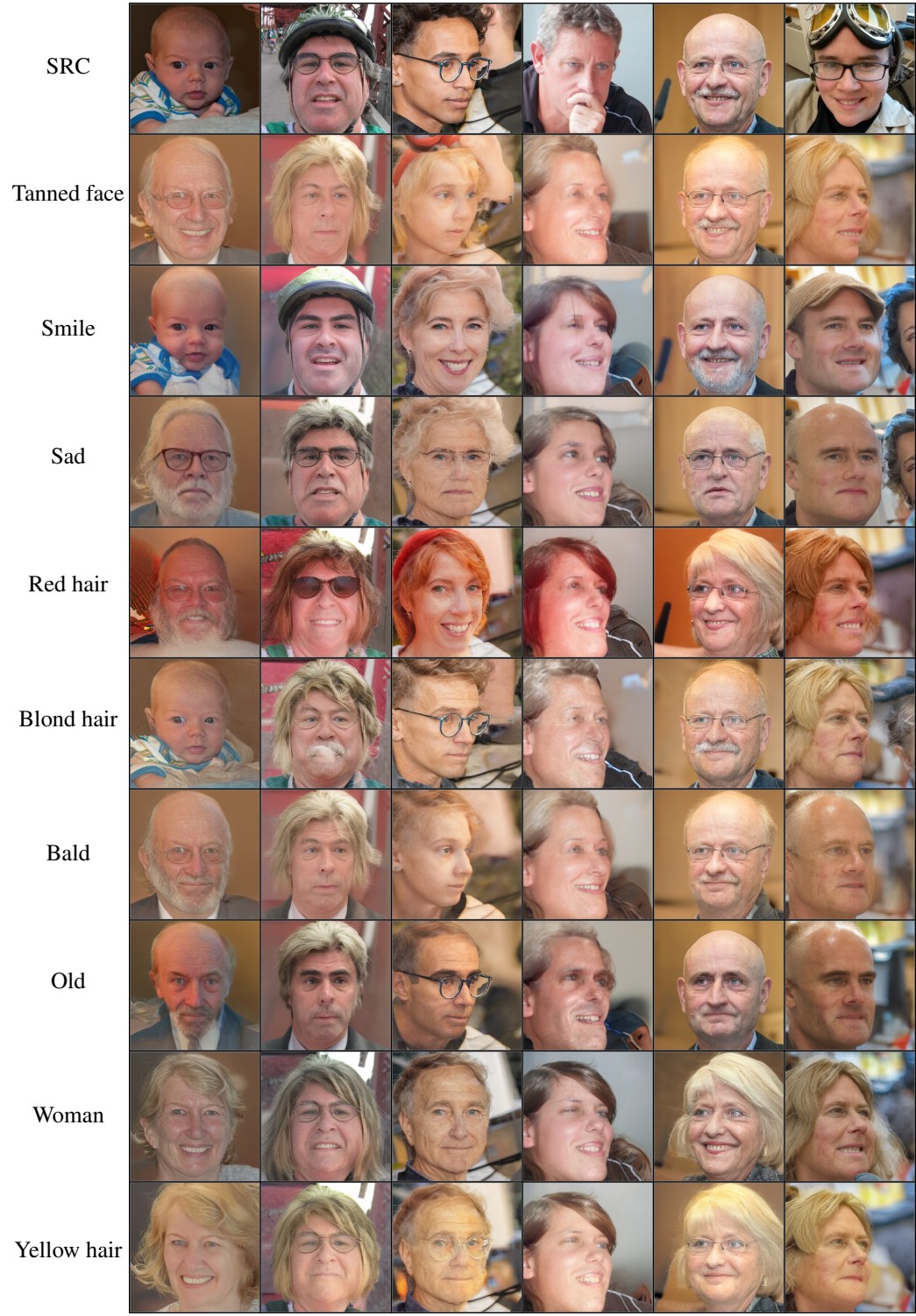

Figure 12: The qualitative results of CLIP-guided DisDiff on FFHQ. The source (SRC) row provides the images of the generated image. The remaining rows of images are generated by shifting the representation along the learned direction of the corresponding factor on FFHQ. DisDiff learns the direction of pure factors unsupervised. The learned factors are listed in the leftmost column. The images are sampled by using the inferred $x_T$.

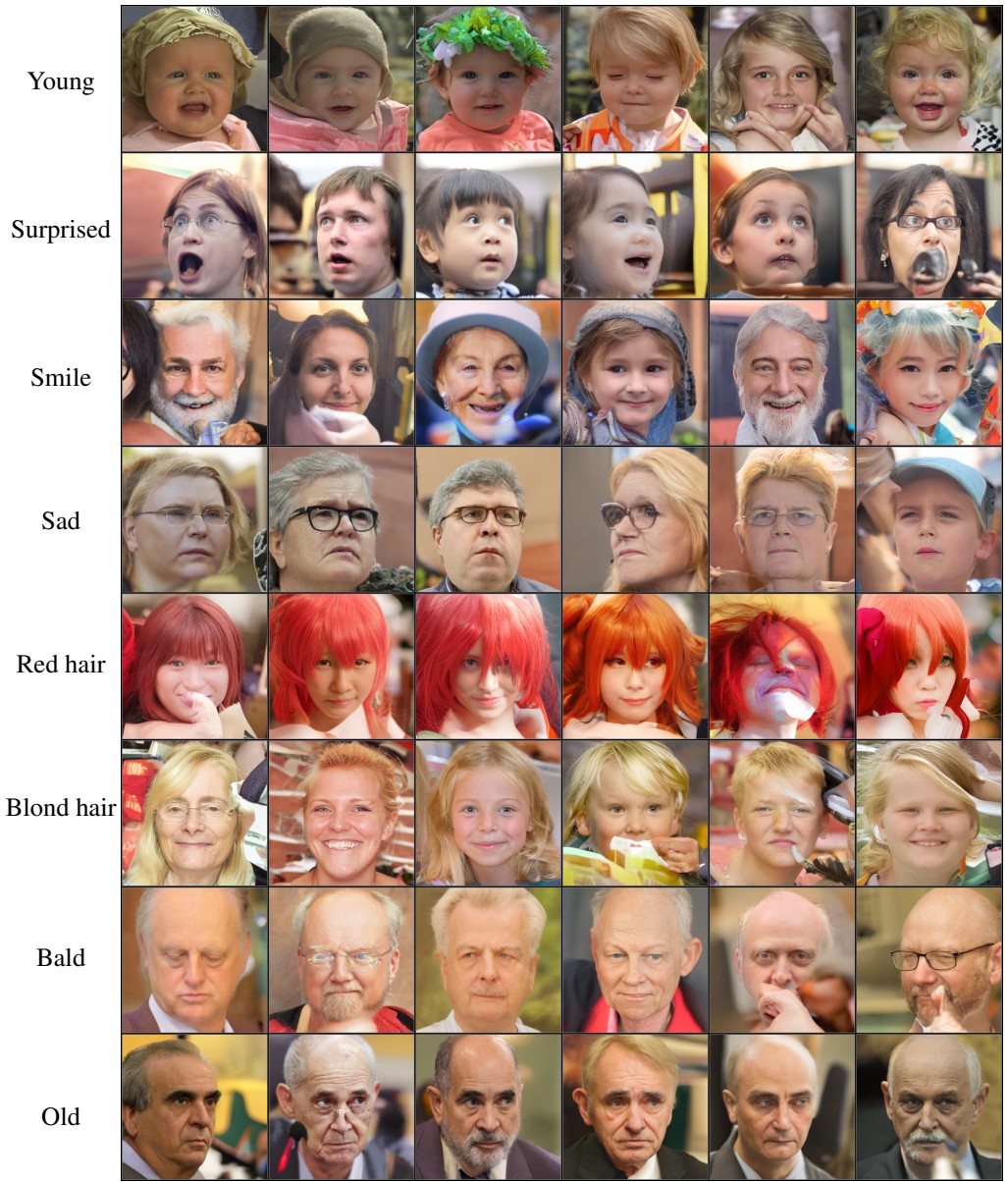

Figure 13: The partial condition sampling results of CLIP-guided DisDiff on FFHQ. The source (SRC) row provides the images of the generated image. The remaining rows of images are generated by shifting the representation along the learned direction of the corresponding factor on FFHQ. DisDiff learns the direction of pure factors unsupervised. The learned factors are listed in the leftmost column. The images are sampled by using the inferred $x_T$.

