# OpenReview forum: "DisDiff: Unsupervised Disentanglement of Diffusion Probabilistic Models"
_NeurIPS.cc/2023/Conference — NeurIPS 2023 poster_

### Official Review · Reviewer_kfVZ · 2023-06-26

**Soundness:** 2 fair
**Presentation:** 2 fair
**Contribution:** 2 fair
**Rating:** 5
**Confidence:** 3

**Summary:**

Imprecisions
- “According to the completeness requirement” add reference
- line 75-76 PADE-> PDAE
- line 270-271. Typo: compare with disco. dissdiff -> compared with disco, disdiff

Strengths
- Achieving disentangled representations in diffusion models is an interesting and useful problem.

Weaknesses
- Missing derivation of reverse diffusion process, Eq 5, 6, 7.
- What is the difference between the proposed model and a latent diffusion model?
- Diff-AE and PDAE should also be considered as baselines in experiments.
- Image generation experiments against Diff-AE and PDAE should be conducted in addition to disentanglement in order to add robustness to the experiments.
- Unclear how Figure 3 is generated. Could you describe the process in detail and explain what you mean by “swapping the representation”.


**Strengths:**

see summary

**Weaknesses:**

see summary

**Questions:**

see summary

**Limitations:**

see summary

---

> ### Author Rebuttal · Authors · 2023-08-09
>
> Thank you for your valuable comments and suggestions. We appreciate your feedback and have made several changes to address your concerns. Please find below our responses to each point you raised.
>
> We have thoroughly revised Section 4 of our paper. We have made significant changes in the presentation and organization of the section, ensuring that our arguments are clearly articulated and well-supported. We highly encourage the reviewers to revisit our revised Section 4, as we are confident that the improvements made will provide a clearer understanding of our research. We appreciate the valuable feedback provided by the reviewers and look forward to receiving further comments that will help us refine and strengthen our paper.
>
> - Imprecisions: We have revised the manuscript according to your suggestions and corrected the imprecisions. We have the typos in the main paper.
> - Derivation of reverse diffusion process (Eq. 5, 6, 7 in our submission): Based on your feedback, we have now included the derivation of these equations.Please refer to comment “Section 4”. Eq.5 is taken from [3] (Please see Equation 5 in it). Eq. 6 is taken from the classifier guidance trick proposed in [1] (Please see Equation 14 in it), the role of it  is to build the score function of conditional distribution.  Eq. 7 is taken from the reverse problem proposed in [2] (Please see Equation 10 in it), the role of it is to recover the data sample with the score function. We believe that this addition helps improve the clarity and completeness of our presentation.
> - Difference between the proposed model and a latent diffusion model: The key difference between our proposed model and a latent diffusion model is that the latter is a general DPM without disentangled representations and the ability to sample according to independent factors. Our method is designed to disentangle general DPMs, providing them with these capabilities. We have clarified this distinction in our revised manuscript.
> - Additional baselines and experiments: We appreciate your suggestion to include Diff-AE and PDAE as baselines in our experiments. We will add these baselines and conducted image generation experiments against them in addition to disentanglement in discussion stage.
> - Clarification on Figure 3: We apologize for any confusion regarding the generation of Figure 3. In our revised manuscript, we have provided a detailed explanation of the process. To "swap the representation," we first encode the images to obtain their representations. We then exchange the representations corresponding to the same factor between two different images. Finally, we decode the swapped representations to generate new images.
>
> We believe that addressing your concerns has significantly improved the quality and clarity of our paper. We appreciate your constructive feedback.
>
> [1] Diffusion Models Beat GANs on Image Synthesis.
>
> [2] Improving Diffusion Models for Inverse Problems using Manifold Constraints. NeurIPS 2022
>
> [3] Unsupervised Representation Learning from Pre-trained Diffusion Probabilistic Models. NeurIPS 2022

---

> > ### Comment · Reviewer_kfVZ · 2023-08-12
> >
> > Thanks for addressing my comments, I updated my score.

---

> > > ### Author Response · Authors · 2023-08-13
> > > **Response**
> > >
> > > Dear Reviewer kfVZ, thank you for the improved rating. We appreciate your valuable feedback and will continue to enhance our work.
> > >
> > > Best regards

---

### Official Review · Reviewer_ZKsn · 2023-07-05

**Soundness:** 2 fair
**Presentation:** 2 fair
**Contribution:** 2 fair
**Rating:** 5
**Confidence:** 2

**Summary:**

This paper proposes to disentangle a pre-trained diffusion probabilistic model (DPM) in an unsupervised way to improve interpretability. The author designed two constraints, invariant condition and variant condition, to guide the disentanglement. The proposed method was evaluated on three synthetic datasets and CelebA, a facial image dataset.

**Strengths:**

- The combination of disentangled representation learning and diffusion-based models is an important and challenging problem.
- Judging solely by the experimental results, the proposed method works to some extent on synthetic datasets.

**Weaknesses:**

- Due to unclear definitions and poor mathematical formulation, I was unable to completely understand the proposed method.
- The author claimed that "the conditional independence of the representations is a necessary condition for disentanglement". However, it is not sufficient. Disentanglement is more than the independence of representations. Even the definition of disentanglement used in this paper is unclear. "For the first time achieving disentangled representation learning in the framework of DPMs" is overclaiming.
- The author's claims are a bit contradictory. The author claimed that "disentangling a DPM can improve the interpretability of DPM" but also admitted that "Our method is completely unsupervised, and without any guidance, the learned disentangled representations on natural image sets may not be easily interpretable by humans." It seems that the author had a goal but failed to achieve it. Is it realistic to consider completely unsupervised disentanglement?

**Questions:**

- What is the mathematical definition of _disentangled conditional sub-gradient fields_? What does "The data distribution $p(x)$ can be **disentangled** into $N$ independent distributions $\\{p(x \mid f^k) \mid k = 1, \dots, N \\}$" mean?
- What does "the data sample space is not **well-organized**" mean?
- Cross entropy compares probability distributions. However, Eq. (10) uses a "distance vector" $d$ and "index" $c$. Eq. (12) even uses a difference of distance vectors. The math here is too questionable to make this paper credible. Please revise this part.

**Limitations:**

Section 6 discussed the limitations of unsupervised disentanglement and diffusion-based methods.

"The potential negative societal impacts are malicious uses." is too general and ambiguous.

---

> ### Author Rebuttal · Authors · 2023-08-09
>
> Thank you for your valuable comments and suggestions. We appreciate your feedback and have made several changes to address your concerns. Please find below our responses to each point you raised.
>
> We have thoroughly revised Section 4 of our paper. We have made significant changes in the presentation and organization of the section, ensuring that our arguments are clearly articulated and well-supported. We highly encourage the reviewers to revisit our revised Section 4, as we are confident that the improvements made will provide a clearer understanding of our research. We appreciate the valuable feedback provided by the reviewers and look forward to receiving further comments that will help us refine and strengthen our paper.
>
>
> - Overclaim: We apologize for any overclaiming in our previous manuscript. We have now revised the manuscript and properly toned down our claims accordingly.
> - Definitions and mathematical formulation: We have carefully revised the manuscript to provide clearer definitions and improved mathematical formulations. Please refer to comment “Section 4”.We believe that these changes have significantly improved the clarity of our paper and helped address your concerns regarding the understanding of the proposed method.
> - Necessary condition for disentanglement: We agree that the conditional independence of the representations is a necessary, but not sufficient, condition for disentanglement. We have revised the manuscript to clarify this point and provide a more accurate statement: "We follow prior works in disentangle representation literature that propose a necessary condition that works in practice for disentanglement. For example, the group constraint in [43], the maximization of mutual information in [23, 41]. We propose to minimize the mutual information between as a necessary condition."
> - Contradictory claims: We apologize for any confusion. We want to demonstrate that：“ Due to the fact that disentangled representation learning on real-world data is quite challenging. While our method demonstrates effectiveness on synthetic datasets, we find that the learned disentangled representations on natural image datasets may not be easily interpretable by humans. We infer that the possible reasons behind it are the requirement of  reconstruction and the weak ability of pretrained DPM. Therefore, we have also included additional results with a more powerful DPM in the appendix that demonstrate the effectiveness of our method for disentangled generation on real-world data. This shows that our method can be beneficial for disentangled generation, We have now revised the manuscript properly and accordingly.
>
> Questions:
> - We apologize for any confusion caused by definitions and mathematical formulation. We have carefully revised the manuscript to provide clearer definitions and improved mathematical formulations. Please refer to the comment “Section 4”. We highly recommend you read that part first, we will clarify the questions proposed.
> - “disentangled conditional sub-gradient fields”: We assume that the dataset is generated by N factors, $D=\lbrace x_0|x_0\sim p(x)\rbrace$, there is a one-one mapping: $h: (f^1,f^2,\dots,f^N)\rightarrow x_0$, there is a set of conditional distribution $\lbrace p(x|f^c)|c = 1,\dots,N\rbrace$. The disentanglement of DPM is to learn the score function  $\nabla_{x_t} \log p(f^c|x_t)$ so that we can sampling from conditional distributions $p(x_t|f^c)$.
> - ”the data sample space is not well-organized“ We mean that the data space is complex, it is difficult to sampling from $p(x|f^c)$, when we have the score function $\nabla_{x_t} \log p(x_t)$, by this we want to emphasize the difficulty of the problem.
> - “Cross entropy”: Please note that the cross entropy here we use is the cross entropy loss function in classification problems. The inputs of this function are “logits” and “labels”(GT). In our setting, we regard the distance as logits and indexes as labels.  As comment “Section 4” shows, we use this cross entropy loss as a tool for implementation of minimizing the distance in the Proposition. Since cross entropy loss is a bounded function, the usage of such a loss function has the benefit for training stability.
>
> We appreciate your constructive feedback, which has helped us improve the quality and clarity of our paper.

---

> > ### Comment · Reviewer_ZKsn · 2023-08-13
> >
> > Thank you for your answers. I appreciate the improved mathematical notation.
> > I'm now slightly more positive about this work, but I still need some time to check the details.
> >
> > A small question: by "one-one mapping", do you mean injection or bijection? Either way, it may be true for synthetic data like 3D Shapes, but it is hardly true for real data because of unannotated factors: there could be multiple examples with the same set of factors. How did you deal with it?
> > (btw, you used $h: x_0 \mapsto (f^1, \dots, f^N)$ in the paper but $h: (f^1, \dots, f^N) \mapsto x_0$ in the rebuttal. Which is correct? Please double-check.)
> >
> > PS: I don't think "cross entropy loss function" is a widely accepted and uniquely defined concept (yes I know how `torch.nn.CrossEntropyLoss` is implemented). When we talk about cross-entropy, it's always defined for probability distributions. Please specify and justify the probabilistic model you want to use.

---

> > > ### Author Response · Authors · 2023-08-14
> > > **Response**
> > >
> > > Dear Reviewer ZKsn,
> > > we would like to thank you for follow-up and increasing the score. We are more than happy for the reviewers to verify the details.  We appreciate your valuable feedback and will continue to enhance our work.
> > >
> > > 1.Regarding the first question
> > >
> > > - We appreciate the reviewer raising this issue. We apologize for our statement not being rigorous enough, as it is tailored to artificial datasets like Shapes3D. What we meant by mapping $h$ to express here is the dependency of $x_0$ on each factor $f^c$, as shown in the probability graph model of Figure 1 (a). The sampled $f^c$ and $x_0$ actually only have a conditional probability dependency, $p(x_0|f^1...f^N)$ (for Shapes3D dataset, $p(x_0|f^1...f^N) = 1$). As we can see from the comment "section 4", we only used this conditional probability (PGM) as our assumption, not the one-one mapping (bijection). Therefore, our method is also effective for general datasets, e.g., CelebA.
> > >
> > > - We will remove the mapping $h$ and modify the sentence to "The data $x_0$ depends on each factor $f^c$, indicating the presence of dependency relationships in the probability graphical model, i.e., $p(x_0|f^1...f^N)$, as illustrated in Figure 1(a)."
> > >
> > > 2.Regarding the second question
> > >
> > > We agree with the reviewer that cross-entropy should be between two distributions, and the probability distribution used should be specified. We would like to change our previous writing to:
> > >
> > > - For Invariant Loss, we use the softmax of the representation distances as the probability $q_I(i=k)$, which indicates the probability of   the largest change of representation occurring in unit $k$ after encoding the conditioned sampled images. The largest change should occur in the $c$-th unit, so for the target distribution $p_I$:  the probability of the $c$-th unit having the largest change is 1, and for all other units it is 0, i.e., $p_I(i=c) = 1, p_I(i=k)=0, k \neq c$.
> > >
> > > - For Variant Loss, we use the softmax of the difference between two representation distances as the probability $q_V(i=k)$, which indicates the probability that  unit $k$ has the smallest representation distance  (consistent with the input image) after encoding the conditioned sampled images. Similarly, we expect the $c$-th unit of representation to be the most consistent. Therefore, for the target probability $p_V$:  the probability of the $c$-th unit being the most consistent is 1, while for all other units, it is 0, i.e., $p_V(i=c) = 1, p_V(i=k)=0, k \neq c$.

---

> > > > ### Comment · Reviewer_ZKsn · 2023-08-15
> > > >
> > > > Thank you for answering my questions and explicitly specifying the probabilistic models. Maybe it's just my personal taste, but I prefer this more rigorous description. I'll check other reviews soon, and I don't have any other questions for now.

---

### Official Review · Reviewer_6mFa · 2023-07-13

**Soundness:** 3 good
**Presentation:** 2 fair
**Contribution:** 3 good
**Rating:** 5
**Confidence:** 5

**Summary:**

This paper proposes a new task of unsupervised disentanglement of diffusion probabilistic models (DPMs) and presents an approach named DisDiff to achieve disentangled representation learning in the framework of DPMs. The authors connect disentangled representation learning to DPMs to take advantage of the remarkable modeling ability of DPMs. DisDiff learns a disentangled representation of the input image in the diffusion process and for each factor, DisDiff learns a disentangled gradient field, which brings new properties for disentanglement literature. The proposed method is evaluated on several real-world datasets, and the results show that DisDiff outperforms state-of-the-art methods in terms of disentanglement quality and interpretability. The paper concludes by discussing potential future directions for applying DisDiff to more general conditioned DPMs and pre-trained conditional DPMs.

**Strengths:**

Strengths:
- Performance: The proposed method is evaluated on several real-world datasets, and the results show that DisDiff outperforms state-of-the-art methods in terms of disentanglement quality and interpretability.

- Clarity: The writing is good overall and the proposed idea in this paper is easy to follow. The authors provide good explanations of the method and the experiments.

- Significance: The proposed method has potential applications in various fields, such as image editing, controllable generation, etc.

- References: The paper provides comprehensive references.

**Weaknesses:**

Weaknesses:
- Overclaim: this paper claims that they are the very first work introducing disentanglement tasks for diffusion probabilistic models (DPM). However, a very recent work [1] in ICML 2023 has introduced and studied this problem. I understand that there might be some timeline issue that makes the authors not aware of this work. But it would be better if the authors could properly cite and discuss this work and revise their claims in the corresponding paragraphs.
- Abuse of notations: there are some wrong notations (might be typos) and abuse of notations, especially in Sec. 4.3. For example, in line 179, it should be $E_\phi^k(\hat{x}_0^c)$ and $E_\phi^k(\hat{x}_0)$. In lines 180 - 181, the conditioned representation should be $E_\phi^c(\hat{x}_0^c)$ and the claim should be that the unconditioned one is closer to $E_\phi^c(\hat{x}_0)$ than the conditioned one. I suggest the authors carefully examine their statements instead of letting the reviewers guess the meaning.
- More baselines: I appreciate that the authors include some of the most important baselines of VAEs and GANs along the line of disentanglement research. However, some more recent diffusion baselines could also be included, e.g., PADE [2] and DiffAE [3] (I understand [1] would be too new to have the open-source codebase to evaluate).
- More metrics for CelebA: there is a tailored quantitative metric [4] for evaluating disentanglement performance on CelebA. It would be better to have the numbers in the tables.
- In line 176, fulfil -> fulfill

[1] Wang, Yingheng, et al. "InfoDiffusion: Representation Learning Using Information Maximizing Diffusion Models." arXiv preprint arXiv:2306.08757 (2023).

[2] Zhang, Zijian, Zhou Zhao, and Zhijie Lin. "Unsupervised representation learning from pre-trained diffusion probabilistic models." Advances in Neural Information Processing Systems 35 (2022): 22117-22130.

[3] Preechakul, Konpat, et al. "Diffusion autoencoders: Toward a meaningful and decodable representation." Proceedings of the IEEE/CVF Conference on Computer Vision and Pattern Recognition. 2022.

[4] Yeats, Eric, et al. "Nashae: Disentangling representations through adversarial covariance minimization." European Conference on Computer Vision. Cham: Springer Nature Switzerland, 2022.

**Questions:**

- Distortion in generated images: although the quantitative metrics (e.g. DCI) are pretty promising, the qualitative results on CelebA still seem a bit distorted with disentangled latent variables. Would there be some training tricks that can be applied to alleviate this issue? Are there any other explanations?
- In line 159, the authors claim that Eq. (7) is derived from Eq. (6) by using Tweedie's formula, however, in my opinion, this is just a straightforward application of reparameterization trick. Can the authors provide the full derivation of the methodology part?

**Limitations:**

There is a section of discussion for limitations.

---

> ### Author Rebuttal · Authors · 2023-08-09
>
> Thank you for your valuable comments and suggestions and the encouraging words. We appreciate the time and effort you took to review our paper. Please find below our responses to your concerns and the changes we have made to address them.
>
>
> We have thoroughly revised Section 4 of our paper. We have made significant changes in the presentation and organization of the section, ensuring that our arguments are clearly articulated and well-supported.  We highly encourage the reviewers to revisit our revised Section 4, as we are confident that the improvements made will provide a clearer understanding of our research. We appreciate the valuable feedback provided by the reviewers and look forward to receiving further comments that will help us refine and strengthen our paper.
>
>
> - Overclaim: We apologize for the oversight in not citing the recent work [1] in ICML 2023. We have now added the reference to this work and revised our claims accordingly in the corresponding paragraphs. We agree that there might have been a timeline issue that led to our unawareness of this work, and we appreciate your understanding. We also acknowledge that our work and [1] can be considered as concurrent works. and we will make this clear in our revised manuscript.
> - Abuse of notations: We apologize for any confusion caused by the notational errors and inconsistencies in our manuscript. We rewrite section 4.3 for better clarity and have carefully revised and corrected these notations. Please refer to the comment “Section 4”. We appreciate your detailed feedback and suggestions, which have helped us improve the clarity of our presentation.
> - More baselines: We agree that including more recent diffusion baselines such as PADE and DiffAE would strengthen our evaluation. We will include these baselines in our experiments and update the results in the paper in the Discussion stage.
> - More metrics for CelebA: We appreciate your suggestion to include the tailored quantitative metric [4] for evaluating disentanglement performance on CelebA. We will add this metric  in the Discussion stage.
>
> For questions:
> - We propose a modified framework for disentangling the DPM of real world data in Appendix G, I and present the results trained on CelebA HQ and FFHQ. We infer  that one possible explanation for the observed distortion is due to the requirement of  reconstruction and the weak ability of pretrained DPM. The disentanglement process might not be so perfectly captured due to the challenge of the complex real world dataset. Another factor could be the choice of loss functions, which might not be optimal for achieving a perfect balance between disentanglement and image quality. We leave it for future work.
>
> - We have modified the draft in Section 4 for clarity. Eq.6 in our submission is taken from the classifier guidance trick proposed in [1] (Please see equation 14 in it), the role of it  is to build the score function of conditional distribution.  Eq.7 in our submission is taken from the reverse problem proposed in [2] (Please See Equation 10 in it), the role of it is to recover the data sample with the score function. We want to use these equations to sample the recovered samples based on representations.
>
> [1] Diffusion Models Beat GANs on Image Synthesis.
>
> [2] Improving Diffusion Models for Inverse Problems using Manifold Constraints. NeurIPS 2022

---

> > ### Comment · Reviewer_6mFa · 2023-08-16
> >
> > Thank you for addressing my comments! The experiments you added look promising. It seems this work achieves considerable improvements on disentanglement tasks. However, I still have some concerns regarding Section 4.
> >
> > - There're still some notations that have not been used properly, e.g. $\mapsto$, which people usually use it to depict how elements from the domain are transformed by the function, not how the entire domain is mapped to a single element in the codomain. There should be clear explanations.
> >
> > - Typos like "follow ..." which should be "following". (Although I appreciate the technical contributions in this paper and the efforts that the authors put into the rebuttal, the paper still needs carefully proofreading and more polishing as a publication.)
> >
> > - Regrading how the authors arrive at the final disentangling loss, I am still a bit confused. It seems the authors introduced some new notations in proposition 1 and the proof sketch (the proof sketch is still not that clear to me though). I think I understood the basic idea of minimizing mutual information. But I am still curious about the full derivation. Could you please provide it?

---

> > > ### Author Response · Authors · 2023-08-16
> > > **Response**
> > >
> > > Dear Reviewer 6mFa,
> > >
> > > Thank you for your valuable comments and for acknowledging the improvements in our work. We genuinely appreciate your feedback. Please find our responses to your concerns below:
> > >
> > >
> > > Regarding the notations in Section 4:
> > >
> > > - We regret that our writing makes you misunderstand the notation here.  The notation here is exactly to depict how elements from the domain are transformed. The input of the function $h$ is an element $(f_1,\dots,f_N)$, which is a vector in $R^N$ space, **NOT** the factor set $\mathcal{C}$. In addition, as shown in our response for Reviewer ZKsn, we will delete the function $h$ and $\mapsto$. We will ensure that the notations are used properly and clearly. We will make the necessary modifications and ensure that the entire domain is mapped correctly in our updated version.
> > >
> > >
> > > - Typos and proofreading:
> > > We appreciate your attention to details. We actually want to use an imperative sentence here.  We will thoroughly proofread the paper to eliminate typos and improve the overall clarity. We are committed to presenting a well-polished publication.
> > >
> > >
> > > Final disentanglement loss "derivation":
> > >
> > > - We appreciate your attention on the proof sketch. Our target of providing proposition 1 is to demonstrate a better understanding of the losses proposed. The loss function and method is derived from an intuitive way as explained in our paper and can not be derived directly from the proposition. Please note that we replace the original notation with new notations for better understanding and clarity rather than introducing new variables in comment “Section 4”, e.g., $\hat{z}^{k|c} = E^k_\phi(\hat{x}^c_0)$.
> > >
> > > - We understand that only a proof sketch may not be sufficiently clear. Due to space constraints and NeurIPS policies (we can not use any url in this page or update our paper now), we present an explanation of the proof sketch and its main ideas, which should help to clarify our proof. If it is possible, we are more than happy for the reviewers to verify the details.
> > >
> > > - We first decompose the mutual information into two terms, entropy and conditional entropy, following the approach in [1,3,4]. Since the encoder is fixed for the disentangling loss, the entropy remains constant. We then introduce a distribution $q$ to divide the conditional entropy into two components: the expectation of $\log q$ and the KL divergence between $p$ and $q$, as also demonstrated in [1,4]. For the first term, considering the PGM in Figure 1(a), we can establish two lemmas to involve $x_0$ in the expectation and resample $\hat{z}^{c|c}$ conditioned on $x_0$ to replace the original $z_c$ in $q$. A similar method to Lemma 1 in [1] can be applied to prove this. Subsequently, we assume $q$ follows a Gaussian distribution and derive the analytical form of the first term. Finally, we can utilize the triangle inequality to obtain the distance of representations as an upper bound. In the second term, we assume that the $p$ distribution follows a Gaussian distribution as well (please note that an assumption on $p$ is necessary to derive an upper bound for MI, as demonstrated in [2,3,4]). By deriving the upper bound of this term, the KL divergence can be expressed as the distance between the means, which corresponds to the distance between the representations.
> > >
> > >
> > > PS: We can also regard the first term and the last term in Eq (6) of "Section 4" as the negative pairs and the middle term  in Eq (6) of "Section 4" as positive pairs in [2], which implies there may be more than one method to provide the proof.
> > >
> > >
> > > [1] InfoGAN: Interpretable Representation Learning by Information Maximizing Generative Adversarial Nets NIPS 2016.
> > >
> > > [2] CLUB: A Contrastive Log-ratio Upper Bound of Mutual Information ICML 2020.
> > >
> > > [3] The im algorithm: A variational approach to information maximization NIPS 2003.
> > >
> > > [4] Deep Variational Bottleneck ICLR 2017.

---

> > > > ### Comment · Reviewer_6mFa · 2023-08-16
> > > >
> > > > Thanks for your reply! I do think adding clear explanations will improve your paper quality a lot!
> > > >
> > > > The last thing is still regarding your proof/derivation. I understood the proof sketch/logic you showed me. What I want is a mathematically formal derivation that starts from probably an ELBO with a mutual information term (you don't have to start with log-likelihood) to how you get the bounds/objectives, by which I can verify the rigorousness of the proposed method. As you said, if you have another way to derive this, I am happy to review that instead of the variational one.
> > > >
> > > > I know there might be difficulties for you to provide a URL/PDF in the official comments. Probably you can use multiple blocks in this thread. I don't think I have any complaints about reviewing this. And I will raise my score if the derivation/proof looks correct to me.

---

> > > > > ### Comment · Area_Chair_8jL4 · 2023-08-18
> > > > >
> > > > > Dear Authors,
> > > > > do you think you can come back to the generous offer of reviewer 6mFA for reviewing the proof / derivation? Since the time frame for discussions  is nearing an end it would be great if reviewer 6mFa can potentially respond?
> > > > > All the best,
> > > > > AC

---

> > > > > ### Author Response · Authors · 2023-08-18
> > > > > **Response**
> > > > >
> > > > > Dear Reviewer 6mFa,
> > > > >
> > > > > Thank you for your patience, and we apologize for the delay in our response as we were double-checking. We greatly appreciate your willingness to review our derivation and your understanding of the potential difficulties in providing a URL or PDF in the official comments. Your dedication to the reviewing process is commendable, and we are truly grateful for your support.
> > > > >
> > > > > We appreciate your open-mindedness to accept alternative derivations, and in the following explanation, we will provide some derivations regarding the mutual information upper bound. Although it may not be such a strict proof, we believe that it will offer a reliable explanation for our method:
> > > > >
> > > > > **Background**: Our encoder transforms the data $x_0$ (hereafter denoted as $x$, the encoder keeps fixed for disentangling loss) into two distinct representations $z^c$ and $z^k$.  We assume that the encoding distribution follows a Gaussian distribution $p(z^c|x)$, with a mean of $E^c(x)$. However, the distribution $p(x|z^c)$ is intractable. We can employ conditional sampling of our DPM $q_\theta(x|z^c)$ to approximate the sampling from distribution $p(x|z^c)$.  Given that we optimize the disentangling loss at each training step, we implement Eq. 6 and Eq.7 in our original manuscript as an efficient alternative to sample from $q_\theta(x|z^c)$. We further assume that the distribution used to approximate $p(z^k|z^c)$ follows a Gaussian distribution $q_\theta(z^k|z^c)$, with a mean of $\hat{z}^{k|c} = E^k(D(z^c))$, where $D$ represents the operations outlined Eq.6 and Eq.7.  We represent the pretrained unconditional DPM using the notation $q_\theta(x) \approx p(x)$.
> > > > >
> > > > > Considering the given context, we can present the PGM related to $q$ as follows:$z^c\rightarrow x\rightarrow z^k$. The first transition is implemented by DPM represented by the distribution $q_\theta$. Subsequently, the second transition is implemented by the encoder, denoted by the distribution $p$.
> > > > >
> > > > > **Proposition 1**: The estimator below is an upper bound of the mutual information $I(z^c, z^k)$. For clarity, we use expectation notation with explicit distribution.
> > > > > $$I_{est} = \mathbb{E}{p(z^c,z^k)} [\log p(z^k|z^c)] -  \mathbb{E}{p(z^c)p(z^k)} [\log p(z^k|z^c)].$$
> > > > > Equality is attained if and only if $z^k$ and $z^c$ are independent.
> > > > >
> > > > > Proof:
> > > > >
> > > > > $$
> > > > > I_{est} - I(z^c, z^k) = \mathbb{E}{p(z^c,z^k)} [\log p(z^k|z^c)] - \mathbb{E}{p(z^c)p(z^k)} [\log p(z^k|z^c)]  - \mathbb{E}{p(z^c,z^k)} [\log p(z^k|z^c) - \log p(z^k)] $$
> > > > > $$
> > > > > = \mathbb{E}{p(z^c,z^k)}[ \log p(z^k)] - \mathbb{E}{p(z^c)p(z^k)} [\log p(z^k|z^c)]
> > > > > $$
> > > > > $$
> > > > > = \mathbb{E}{p(z^k)}[ \log p(z^k) - \mathbb{E}{p(z^c)} [\log p(z^k|z^c)]].
> > > > > $$
> > > > > Using Jensen’s Inequality, we obtain $\log p(z^k) = \log(\mathbb{E}{p(z^c)}[p(z^k|z^c)]) \geqslant \mathbb{E}{p(z^c)}[\log p(z^k|z^c)]$. Therefore, the following holds:
> > > > > $$
> > > > > I_{est} - I(z^c, z^k) = \mathbb{E}{p(z^k)}[ \log p(z^k) - \mathbb{E}{p(z^c)} [\log p(z^k|z^c)]]
> > > > > $$
> > > > > $$
> > > > > I_{est} - I(z^c, z^k) \geqslant 0.
> > > > > $$
> > > > >
> > > > > Since the distribution $p(z^k|z^c)$ is intractable, we employ $q_\theta(z^k|z^c)$ to approximate $p(z^k|z^c)$.
> > > > > $$
> > > > > I_\theta = \mathbb{E}{p(z^c,z^k)}[\log q_\theta(z^k|z^c)] - \mathbb{E}{p(z^c)p(z^k)}[\log q_\theta(z^k|z^c)].
> > > > > $$
> > > > >
> > > > > **Proposition 2**: The new estimator can be expressed as follows:
> > > > > $$
> > > > > I_\theta = -\mathbb{E}p(z^c, z^k)[\|z^k-\hat{z}^{k|c}\|] + \mathbb{E}{p(z^c)}\mathbb{E}{q_\theta(\hat{x})}\mathbb{E}{p(\hat{z}^k|\hat{x})}[\|\hat{z}^k-\hat{z}^{k|c}\|] + C_1.
> > > > > $$
> > > > > The first term corresponds to $d^p_k$ in Eq 11 of the original manuscript, while the second term corresponds to Eq. 9 in the original manuscript.
> > > > >
> > > > > Proof:
> > > > >
> > > > > Considering the first term, we obtain:
> > > > > $$
> > > > > \log q_\theta(z^k|z^c) = \log \exp(-\|z^k - E^k(D(z^c))\|) + C_2 = -\|z^k - \hat{z}^{k|c}\| + C_2.
> > > > > $$
> > > > >
> > > > > For the second term, we obtain:
> > > > > $$
> > > > > \mathbb{E}{p(z^k)}[\log q_\theta(z^k|z^c)] = \int \int p(x,z^k)\log q_\theta(z^k|z^c) dz^kdx
> > > > > $$
> > > > > $$
> > > > > = \int p(x)\int p(z^k|x)\log q_\theta(z^k|z^c) dz^kdx \approx \int q_\theta(\hat{x})\int p(\hat{z}^k|\hat{x})\log q_\theta(\hat{z}^k|z^c) d\hat{z}^kd\hat{x}
> > > > > $$
> > > > >
> > > > > $$
> > > > > =  \mathbb{E}{q_\theta(\hat{x})}\mathbb{E}{p(\hat{z}^k|\hat{x})}[\log q_\theta(\hat{z}^k|z^c)] = \mathbb{E}{q_\theta(\hat{x})}\mathbb{E}{p(\hat{z}^k|\hat{x})}[\|\hat{z}^k-E^k(D(z^c))\|]  + C_3= \mathbb{E}{q_\theta(\hat{x})}\mathbb{E}{p(\hat{z}^k|\hat{x})}[\|\hat{z}^k-\hat{z}^{k|c}\|] + C_3.
> > > > > $$

---

> > > > > ### Author Response · Authors · 2023-08-18
> > > > > **Response Part 2:**
> > > > >
> > > > > **Proposition 3**: If the following condition is fulfilled, the new estimator $I_{\theta}$ can serve an upper bound for the mutual information $I(z^c,z^k)$, i.e., $I_{\theta} \geqslant  I(z^c, z^k)$. The equality holds when $z^k$ and $z^c$ are independent. We denote  $q_\theta(z^k|z^c)p(z^c)$ as $q_\theta(z^k, z^c)$.
> > > > >
> > > > > $$
> > > > > D_{KL}(p(z^k,z^c)|| q_\theta(z^k, z^c)) \leqslant D_{KL}(p(z^k)p(z^c)|| q_\theta(z^k, z^c)).
> > > > > $$
> > > > >
> > > > > Proof:
> > > > >
> > > > > $$
> > > > > I_{\theta} - I(z^c, z^k) =  \mathbb{E}{p(z^c,z^k)}[\log q_\theta(z^k|z^c)] - \mathbb{E}{p(z^c)p(z^k)}[\log q_\theta(z^k|z^c)] - \mathbb{E}{p(z^c,z^k)} [\log p(z^k|z^c) - \log p(z^k)]
> > > > > $$
> > > > >
> > > > > $$
> > > > > = \mathbb{E}{p(z^c,z^k)}[\log q_\theta(z^k|z^c) - \log p(z^k|z^c)] + \mathbb{E}{p(z^c)p(z^k)}[-\log q_\theta(z^k|z^c) + \log p(z^k)]
> > > > > $$
> > > > >
> > > > > $$
> > > > > = \mathbb{E}{p(z^c)p(z^k)}[-\log q_\theta(z^k|z^c) + \log p(z^k)] -  \mathbb{E}{p(z^c,z^k)}[\log p(z^k|z^c) - \log q_\theta(z^k|z^c)]
> > > > > $$
> > > > >
> > > > > $$
> > > > > = \mathbb{E}{p(z^c)p(z^k)}\left[\log \frac{p(z^k)p(z^c)}{q_\theta(z^k|z^c)p(z^c)}\right] -  \mathbb{E}{p(z^c,z^k)}\left[\log \frac{p(z^k|z^c)p(z^c)}{q_\theta(z^k|z^c)p(z^c)}\right]
> > > > > $$
> > > > >
> > > > > $$
> > > > > = D_{KL}(p(z^c)p(z^k)||q_\theta(z^k,z^c)) - D_{KL}(p(z^c, z^k)||q_\theta(z^k,z^c)).
> > > > > $$
> > > > > If $D_{KL}(p(z^k,z^c)|| q_\theta(z^k, z^c)) \leqslant D_{KL}(p(z^k)p(z^c)|| q_\theta(z^k, z^c))$, we obtain:
> > > > > $$
> > > > > I_{\theta}\geqslant I(z^c, z^k).
> > > > > $$
> > > > > Since the condition is not yet applicable, we will continue exploring alternative conditions.
> > > > >
> > > > > **Proposition 4**: If $D_{KL}(p(z^k|z^c)||q_\theta(z^k|z^c))\leqslant \epsilon$, then, either of the following conditions holds: $I_{\theta}\geqslant I(z^c,z^k)$ or $\|I_{\theta}- I(z^c,z^k)\| \leqslant \epsilon$.
> > > > >
> > > > > Proof:
> > > > > If $D_{KL}(p(z^k,z^c)|| q_\theta(z^k, z^c)) \leqslant D_{KL}(p(z^k)p(z^c)|| q_\theta(z^k, z^c))$, then
> > > > > $$
> > > > > I_{\theta}\geqslant I(z^c,z^k),
> > > > > $$
> > > > > else:
> > > > > $$
> > > > > D_{KL}(p(z^k)p(z^c)|| q_\theta(z^k, z^c)) < D_{KL}(p(z^k,z^c)|| q_\theta(z^k, z^c)) =\mathbb{E}{p(z^c,z^k)}\left[\log \frac{p(z^k|z^c)}{q_\theta(z^k|z^c)}\right] = D_{KL}(p(z^k|z^c)||q_\theta(z^k|z^c))\leqslant \epsilon.
> > > > > $$
> > > > > $$
> > > > > \|I_\theta - I(z^c,z^k)\|<\max(D_{KL}(p(z^k,z^c)|| q_\theta(z^k, z^c)), D_{KL}(p(z^k)p(z^c)|| q_\theta(z^k, z^c))) \leqslant \epsilon.
> > > > > $$
> > > > > **Proposition 5**:
> > > > > Consider a PGM $z^c\rightarrow x\rightarrow z^k$; if $D_{KL}(q_\theta(x|z^c)|p(x|z^c)) = 0$, it follows that $D_{KL}(p(z^k|z^c)||q_\theta(z^k|z^c)) = 0$.
> > > > >
> > > > > Proof:
> > > > > if $D_{KL}(q_\theta(x|z^c)|p(x|z^c)) = 0$, we have $q_\theta(x|z^c) = p(x|z^c)$.
> > > > > $$
> > > > > D_{KL}(p(z^k|z^c)||q_\theta(z^k|z^c)) = \mathbb{E}{p(z^c,z^k)}\left[\log \frac{p(z^k|z^c)}{q_\theta(z^k|z^c)}\right] = \mathbb{E}{p(z^c,z^k)}\left[\log \frac{\int p(x|z^c)p(z^k|x)dx}{\int q_\theta(x|z^c)p(z^k|x)dx}\right] = \mathbb{E}{p(z^c,z^k)}[\log 1] = 0.
> > > > > $$
> > > > >
> > > > > Ideally, by minimizing $D_{KL}(q_\theta(x|z^c)|p(x|z^c))$ to 0, we can establish $I_\theta$ either as an upper bound or as an estimator of $I(z^c,z^k)$.
> > > > >
> > > > > **Proposition 6**: we can minimize $D_{KL}(q_\theta(x|z^c)|p(x|z^c))$ by minimizing the following objective:
> > > > > $$
> > > > > \mathbb{E}p(z^c)\mathbb{E}q_\theta(x|z^c) [\|z^c - \hat{z}^{c|c}\|] + C_4.
> > > > > $$
> > > > > Note that the equation above exactly corresponds to $d^p_c$ in Eq. 11,  and $d^n_k$ is a constant.
> > > > >
> > > > > Proof:
> > > > >
> > > > > $$
> > > > > D_{KL}(q_\theta(x|z^c)|p(x|z^c)) = \int p(z^c) \int q_\theta(x|z^c)\log \frac{q_\theta(x|z^c)}{p(x|z^c)}dxdz^c.
> > > > > $$
> > > > > We focus on the following term:
> > > > > $$
> > > > > \int q_\theta(x|z^c)\log \frac{q_\theta(x|z^c)}{p(x|z^c)}dx = \int q_\theta(x|z^c)[\log q_\theta(x|z^c) - \log p(x|z^c)]dx.
> > > > > $$
> > > > > Applying the Bayes rule, we obtain:
> > > > > $$
> > > > > = \int q_\theta(x|z^c)[\log q_\theta(x|z^c) - \log \frac{p(z^c|x)p(x)}{p(z^c)}]dx
> > > > > $$
> > > > >
> > > > > $$
> > > > > = \int q_\theta(x|z^c)\left[\log \frac{q_\theta(x|z^c)}{p(x)} - \log p(z^c|x) + \log p(z^c)\right]dx
> > > > > $$
> > > > >
> > > > > $$
> > > > > = D_{KL}(q_\theta(x|z^c)|p(x)) - \mathbb{E}q_\theta(x|z^c)[\log p(z^c|x)] + \log p(z^c)
> > > > > $$
> > > > > $$
> > > > > \leqslant - \mathbb{E}q_\theta(x|z^c)[\log p(z^c|x)] + \log p(z^c)
> > > > > $$
> > > > >
> > > > > $$
> > > > > = - \mathbb{E}q_\theta(x|z^c)[\log \exp(-\|z^c-E^c(x)\|)] + \log p(z^c) + C_5
> > > > > $$
> > > > >
> > > > > $$
> > > > > = \mathbb{E}q_\theta(x|z^c)[\|z^c-\hat{z}^{c|c}\|] + \log p(z^c) + C_5.
> > > > > $$
> > > > >
> > > > > We can minimize $\mathbb{E}p(z^c)\mathbb{E}q_\theta(x|z^c)[\|z^c-\hat{z}^{c|c}\|]$ instead.
> > > > >
> > > > > Note that the loss usually cannot be optimized to a global minimum in a practical setting, but  how the gap influences $I_\theta$ is beyond the scope of this paper. We leave it as future work. We have primarily drawn upon the proofs in [1] and VAE [2] as key literature sources to prove our claims.  As shown in Figure 4, our model has successfully fitted the distribution of $p(x|z^c)$ using $q_\theta(x|z^c)$. Therefore, we can understand that model's loss can be regarded as $I_{est}$.
> > > > >
> > > > > [1] CLUB: A Contrastive Log-ratio Upper Bound of Mutual Information. ICML 2020.
> > > > >
> > > > > [2] Auto-Encoding Variational Bayes. ICLR 2014
> > > > >
> > > > >
> > > > >
> > > > > Once again, thank you for your invaluable feedback and your willingness to review our derivation. We look forward to receiving your comments and will gladly address any concerns you may have. Regardless of your final evaluation, we are grateful for your assistance and dedication to the review process.

---

> ### Comment · Area_Chair_8jL4 · 2023-08-20
>
> Dear Reviewer 6mFa,
>
> thanks a lot for your significant discussion of this work and your community service! The authors have now added significant additional material as a response and I wanted to ask if you are satisfied with their answer given that the discussion period is soon coming to an end.
>
> All the best,
>
> Your AC.

---

> > ### Comment · Reviewer_6mFa · 2023-08-20
> >
> > Dear Authors,
> >
> > Thank you for providing this detailed proof promptly! I understand the discussion deadline is coming soon. So I will only raise two questions regarding the proof.
> >
> > - In Proposition 4, the second case seems to state that $I_\theta$ will be a lower bound instead of an upper bound.
> >
> > - How does the "triangular" objective in the proof sketch arrive according to the propositions you give?

---

> > > ### Author Response · Authors · 2023-08-21
> > > **Response**
> > >
> > > Dear Reviewer 6mFa,
> > >
> > > Thank you for your insightful questions. We would like to address your concerns as follows:
> > >
> > >
> > > - Regarding Proposition 4, the second case of proposition 4 demonstrates that our estimator can serve as an estimator $I_\theta$ with an absolute error bounded by a small positive number $\epsilon$. Recall that our goal is to optimize mutual information. It should be more beneficial to optimize an estimated mutual information rather than just its upper bound. Therefore, when our estimator serves its purpose, whether it acts as an upper or lower bound becomes less critical. We appreciate your feedback on this point and will update our comments and the manuscript to make this point clearer in the updated version.
> > >
> > >
> > > - As for the "triangular" objective in the proof sketch, we have already optimized the original proof to obtain a more clear and simplified one. We are now providing an updated version without it which can provide more clear explanations for our method.
> > >
> > >
> > > We hope that these clarifications address your concerns and help you better understand our work. Please let us know if you have any further questions or require additional information.

---

> > > > ### Comment · Reviewer_6mFa · 2023-08-21
> > > >
> > > > Thank you for the clarification. I think most of my comments have been properly addressed. I will raise the score accordingly. However, I recommend the authors to add those proof in their paper/appendix for better illustrations. And the paper should be further polished (carefully and thoroughly) for publishing.

---

> > > > > ### Author Response · Authors · 2023-08-21
> > > > > **Response**
> > > > >
> > > > > Dear Reviewer 6mFa,
> > > > >
> > > > > Thank you for your positive feedback and for raising the score. We truly appreciate the time and effort you have dedicated to reviewing our work and the proof.
> > > > >
> > > > > We acknowledge your recommendation to include the proof in the paper/appendix, and we will adopt this recommendation. Furthermore, we assure you that we will meticulously and thoroughly polishing the final version of our paper, including the appendix, to ensure the presentation of high quality.

---

### Author Rebuttal · Authors · 2023-08-09

# Section 4
We assume that a dataset $\mathcal{D} = \lbrace x_0\sim p(x_0)\rbrace$ is generated by $N$ underlying ground truth factors $\mathcal{C} =\lbrace f^c|c=1,\dots,N\rbrace$, where $p(x_0)$ is the data distribution, i.e., each sample is generated by the underlining factors, $h: (f^1, \dots, f^N) \mapsto x_0, \forall x_0 \in \mathcal{D}$ and each factor $f^c \sim p(f^c)$, where $p(f^c)$ is the distribution of factor $c$. The conditional distributions of the factors are $\lbrace p(x_0|f^c)|c=1,\dots,N\ \rbrace$, each of which can be shown as a curved surface in Fig. A (b).
The relation between the conditional distribution and data distribution can be formulated as:
$$
    p(x_0) = \int p(x_0|f^c)p(f^c)d f^c, (1)
$$
Note that $f^1,\dots,f^N$ and $x_0$ form a v-structure in PGM, as shown in Fig. A (a). A DPM learns a model $\epsilon_\theta(x_t,t)$ to predict the noise added to a sample $x_t$, then this can be used to derive a score function: $\nabla_{x_t}\log p(x_t) = {1/ \sqrt{\alpha_t - 1}}\cdot\epsilon_\theta(x_t,t)$. Follow [6] to use Bayes Rule, we can write the score function of the conditional distribution as:
$$
    \nabla_{x_t}\log p(x_t|f^c) = \nabla_{x_t}\log p(f^c|x_t) + \nabla_{x_t}\log p(x_t), (2)
$$
By using a score function of $p(x_t|f^c)$, we can sampling data conditioned on the factor $f^c$.
In addition, Eq. 2 can be extended to one conditioned on $\mathcal{S} \subseteq \mathcal{C}$ by replacing $f^c$ with $S$.
The goal of disentangling a DPM is to model $\nabla_{x_t} \log p(x_t|f^c)$ for each factor $c$.
Base on Eq. 2, we can learn $\nabla_{x_t}\log p(f^c|x_t)$ for a pre-trained DPM. However, $f^c$ is unknown in an unsupervised setting. Fortunately, we can learn a set of  disentangled representations $\lbrace z^c|   c=1,\dots,N\brace$.
There are two requirements for these representations: $(i)$ They include all information of $x_0$,  named as *completeness*, and $(ii)$ there exists one-one mappings: $z^c \mapsto f^c$ for each factor $c$, named as *disentanglement*. With these representations, we can use $\nabla_{x_t}\log p(z^c|x_t)$ to approximate $\nabla_{x_t}\log p(f^c|x_t)$.

Our target is to disentangle a pre-trained unconditional DPM on dataset $\mathcal{D}$: $p_\theta(x_{t-1}|x_t) = \mathcal{N} (x_{t-1};\mu_\theta(x_t, t), \sigma_t)$ in an unsupervised manner.
Specifically, given $x_0\in\mathcal{D}$, for each factor $c\in\mathcal{C}$, we can learn the disentangled representation $z^c$ via an encoder $E_\phi$, and the corresponding disentangled gradient field $\nabla_{x_t} \log p(z^c|x_t)$. Since random variables $z^1,\dots,z^N$ and $x_0$ forms a common cause structure, $z^1,\dots,z^N$ are independent when conditioned on $x_0$. We prove that this is also hold for $x_t$. We thus formulate the gradient field $\nabla_{x_t} \log p(z^S|x_t)$, $z^{\mathcal{S}} = \lbrace z^c|f^c \in \mathcal{S}\rbrace$ conditioned on subset $\mathcal{S} \subseteq \mathcal{C}$ as
$$
     \nabla_{x_t} \log p(z^S|x_t) = \sum_{f^c\in S}\nabla_{x_t} \log p(z^c|x_t). (3)
$$
We follow [6,41], to model the conditional reverse process as a Gaussian distribution $p_\theta(x_{t-1}|x_t, z^{\mathcal{S}})$. Together with Eq. 3, the distribution has a shifted mean given by:
$$
     \mathcal{N} (x_{t-1}; \mu_\theta(x_t, t) + \sigma_t\sum_{f^c\in \mathcal{S}}\nabla_{x_t} \log p(z^c|x_t), \sigma_t). (4)
$$
Directly using $\nabla_{x_t} \log p(z^c|x_t)$ complicates the diffusion model training pipeline [14]. We thus use a decoder $G_\psi(x_t, z^c, t), f^c\in \mathcal{S}$ to estimate the gradient fields $\nabla_{x_t} \log p(z^\mathcal{C}|x_t)$. To achieve this goal, we first use $\sum_{f^c\in \mathcal{C}}G_\psi(x_t,z^c,t)$ to estimate $\nabla_{x_t} \log p(z^\mathcal{S}|x_t)$ based on Eq.3.
Therefore, we adopt the loss inPDAE [41] but using summation of $G_\psi(x_t,z^c,t)$ instead as
$$
L_r  = \mathop{\mathbb{E}}\limits_{x_0,t,\epsilon}\|\epsilon - \epsilon_\theta(x_t,t)
          + \frac{\sqrt{\alpha_t}\sqrt{1-\bar \alpha_t}}{\beta_t}\sigma_t\sum_{f^c\in \mathcal{C}}G_\psi(x_t,z^c,t)\|. (5)
$$

The above equation can result in that $x_0$ can be reconstructed using all disentangled representations $E_\phi(x_0)$, which is exactly the *completeness* requirement of disentanglement. Secondly, to meet the *disentanglement* requirement, each $G_\psi(x_t,z^c,t)$ should approximate $\nabla_{x_t} \log p(z^c|x_t)$, individually. With a learnt $G_\psi(x_t,z^c,t)$, we can sample $x^c_0$ conditioned on the disentangled representation $z^c$.


We introduce the Disentangling Loss for the *disentanglement*, and to constrain that $G_\psi(x_t,z^c,t)$ is an estimation of $\nabla_{x_t} \log p(z^c|x_t)$.
Since it is hard to find the sufficient condition for disentanglement in unsupervised setting, we follow prior works that propose necessary conditions worked in practice, e.g., the group constraint in [40], the maximization of mutual information in [22,1 in 6mFa].
We propose to minimize the mutual information between $z^c$ and $z^k$, $k \neq c$, as a necessary condition to disentangle a DPM.

In the following, we transfer the minimizing of mutual information to the constraints in the representation space.
We denote $\hat x_0$ is sampled from the pre-trained unconditioned DPM using $\epsilon_\theta(x_t,t)$, $\hat x^c_0$ is conditioned on $z^c$ and sampled using $G_\psi(x_t,z^c,t)$.
Then we can extract the representations from the samples $\hat x_0$ and $\hat x^c_0$ with $E_\phi$ as $\hat z^k = E_\phi^k(\hat x_0)$ and $\hat z^{k|c} = E_\phi^k(\hat x^c_0)$, respectively.
According to Proposition 1 (Proof sketch in Fig. B), we can minimize an proposed variational upper bound to minimize the mutual information.



**Proposition 1** For a PGM in Fig. A (a), an upper bound of the mutual information between $z^c$ and $z^k$ (where $k \neq c$) is as following:
$$
    \min \mathop{\mathbb{E}}\limits_{k,c,x_0,\hat{x}_0^c}
    \|\hat{z}^{k|c} - \hat{z}^k\|- \|\hat{z}^{c|c} - \hat{z}^c\| +
    \|z^c - \hat{z}^{c|c}\| (6)
$$

---

### Author Response · Authors · 2023-08-11
**More Baseline Results Added (PADE [2] and DiffAE [3])**

We would like to thank the reviewers (6mFa, kfVZ) for their suggestion to include a comparison with more recent diffusion baselines, such as PADE [2] and DiffAE [3]. We are happy to provide the requested comparison, and our experimental results are as follows:

| Model             | SSIM($\uparrow$)         | LPIPS($\downarrow$) |  MSE($\downarrow$)  | DCI($\uparrow$)   | Factor VAE($\uparrow$)   |
|-------------------|----------------|---------------|---------------|---------------|---------------|
|PDAE | 0.9830 | 0.0033 | 0.0005 | 0.3702 | 0.6514 |
| DiffAE | **0.9898** | 0.0015 | **8.1983e-05** | 0.0653 | 0.1744 |
| DisDiff (ours)   | 0.9484 | **0.0006** | 9.9293e-05 | **0.723** | **0.902** |
-------------------

The relatively poor disentanglement performance of the two baselines can be attributed to the fact that neither of these models was specifically designed for disentangled representation learning, and thus, they generally do not possess such capabilities. DiffAE primarily aims to transform the diffusion model into an autoencoder structure, while PDAE seeks to learn an autoencoder for a pre-trained diffusion model. Unsupervised disentangled representation learning is beyond their scope.

Traditional disentangled representation learning often faces a trade-off between generation quality and disentanglement ability (e.g., betaVAE [1], betaTCVAE [2], FactorVAE [3]). However, fortunately, disentanglement in our model does not lead to a significant drop in generation quality; in fact, some metrics even show improvement. The reason for this, as described in DisCo [4], is that using a well pre-trained generative model helps avoid this trade-off.

[1] Higgins I, Matthey L, Pal A, et al. beta-vae: Learning basic visual concepts with a constrained variational framework[C]//International conference on learning representations. 2016.

[2] Chen R T Q, Li X, Grosse R B, et al. Isolating sources of disentanglement in variational autoencoders[J]. Advances in neural information processing systems, 2018, 31.

[3] Kim H, Mnih A. Disentangling by factorising[C]//International Conference on Machine Learning. PMLR, 2018: 2649-2658.

[4] Learning Disentangled Representation by Exploiting Pretrained Generative Models: A Contrastive Learning View

---

### Author Response · Authors · 2023-08-14
**Results of More Metrics for CelebA**

We would like to thank reviewer 6mFa for pointing out the tailored metric [1] for evaluating performance on the CelebA dataset. We appreciate the valuable suggestions and believe that incorporating the metric for real-world data into our paper makes it more convincing.

Following reviewer 6mFa’s advice, we have adopted the TAD and FID metrics from [1] to measure the disentanglement and generation capabilities of DisDiff, respectively. As shown in the table below, DisDiff still outperforms new baselines in comparison. It is worth mentioning that InfoDiffusion [1] has made significant improvements in both disentanglement and generation capabilities on CelebA compared to previous baselines. This has resulted in DisDiff's advantage over the baselines on CelebA being less pronounced than on Shapes3D.


| Model             | TAD         | FID  |
|-------------------|----------------|---------------|
| $\beta$-VAE | 0.088 ± 0.043 | 99.8 ± 2.4 |
| InfoVAE | 0.000 ± 0.000 | 77.8 ± 1.6   |
| Diffae | 0.155 ± 0.010 | 22.7 ± 2.1  |
| InfoDiffusion| 0.299±0.006 | 23.6 ± 1.3   |
| DisDiff (ours)   | **0.3054 ± 0.010** | **18.249 ± 2.1** |

The results are taken from Table 2 in [1], except for the results of DisDiff. For TAD, we follow the method used in [1] to compute the metric. We also evaluate the metric for 5-folds. For FID, we also follow [1] to compute the metric with five random sample sets of 10,000 images. Specifically, we use the official github repo (ericyeats/nashae-beamsynthesis) of [2] to compute TAD, and the official github repo (mseitzer/pytorch-fid) of [3] to compute FID.

[1] InfoDiffusion: Representation Learning Using Information Maximizing Diffusion Models. in ICML 2023.

[2] Nashae: Disentangling representations through adversarial covariance minimization. in ECCV 2022

[3] GANs Trained by a Two Time-Scale Update Rule Converge to a Local Nash Equilibrium. in NeurIPS 2017.

---

### Decision · Program_Chairs · 2023-09-21

**Decision:**

Accept (poster)

**Comment:**

The authors performed additional experiments supporting their work and significantly extended the framework which should be of interest to the wider NeurIPS community.

From the discussion with the reviewers there are no open or controversial points left. The authors exploit an inductive bias which is unique to diffusion models and thus not available in classical approaches. While the specific use and introduction of the inductive bias could have been considered incremental and maybe straightforward, the large scale empirical evaluation on both synthetic and real world datasets are significant contributions and during the rebuttal the authors provided significantly more evidence for their framework (in particular in discussion with reviewer 6mFA). Among these baselines only a more advanced VAE based method joint with more metrics for disentanglement seem missing. For example the considered baselines like beta-VAE are indeed well established but already quite old. While beta-VAE may struggle with the mentioned generation vs. disentanglement tradeoff due to the additional regularization, more modern VAE based methods will not have this problem. However as pointed out by the authors even modern VAE based approaches would not be able to exploit the proposed inductive bias which is specific for diffusion based models.

While this is overall a borderline paper, the changes deemed necessary from the reviewers perspective seem feasible without an additional iteration and I would thus recommend acceptance while assuming that the authors will keep their word of implementing the proposed changes.